# Oxytocin, Vasopressin and Stress: A Hormetic Perspective

**DOI:** 10.3390/cimb47080632

**Published:** 2025-08-07

**Authors:** Hans P. Nazarloo, Marcy A. Kingsbury, Hannah Lamont, Caitlin V. Dale, Parmida Nazarloo, John M. Davis, Eric C. Porges, Steven P. Cuffe, C. Sue Carter

**Affiliations:** 1Department of Psychiatry, College of Medicine-Jacksonville, University of Florida, 580 West 8th St., Tower II, 6th Floor, Jacksonville, FL 32209, USA; steven.cuffe@jax.ufl.edu (S.P.C.); suecarterporges@gmail.com (C.S.C.); 2Lurie Center for Autism, Mass General Research Institute, Harvard Medical School, Charlestown, MA 02129, USA; makingsbury@mgh.harvard.edu (M.A.K.); hhl23@gsbs.rutgers.edu (H.L.); 3Dedman College of Humanities and Sciences, Southern Methodist University, Dallas, TX 75205, USA; cate@smu.edu; 4Traumatic Stress Research Consortium, Kinsey Institute, Indiana University, Bloomington, IN 47405, USA; pnazarl@iu.edu; 5Department of Psychiatry, University of Illinois at Chicago, Chicago, IL 60607, USA; davisjm@uic.edu; 6Department of Clinical and Health Psychology, College of Public Health and Health Professions, University of Florida, Gainesville, FL 32611, USA; eporges@ufl.edu; 7Center for Cognitive Aging and Memory, Evelyn F. and William L. McKnight Brain Institute, University of Florida, Gainesville, FL 32611, USA

**Keywords:** oxytocin, vasopressin, stress, sociality, hormesis

## Abstract

The purpose of this article is to examine a previously unrecognized role for the vasopressin–oxytocin (VP-OT) system in mammalian “stress-response hormesis.” The current review adds hormesis to the long list of beneficial effects of OT. Hormesis, a biphasic adaptive response to low-level stressors, is introduced here to contextualize the dynamic roles of oxytocin and vasopressin. As with hormesis, the properties of the VP-OT system are context-, time-, and dose-sensitive. Here we suggest that one key to understanding hormesis is the fact that VP and OT and their receptors function as an integrated system. The VP-OT system is capable of changing and adapting to challenges over time, including challenges necessary for survival, reproduction and sociality. Prior research suggests that many beneficial effects of OT are most apparent only following stressful experiences, possibly reflecting interactions with VP, its receptors and other components of the hypothalamic–pituitary–adrenal axis. The release of OT is documented following various kinds of hormetic experiences such as birth, vigorous exercise, ischemic events and the ingestion of emetics, including psychedelics. The phasic or cyclic modulation of VP and related “stress” hormones, accompanied or followed by the release of OT, creates conditions that conform to the core principles of hormesis. This concept is reviewed here in the context of other hormones including corticotropin releasing hormone (CRH) and urocortin, as well as cytokines. In general, VP and classic “stress hormones” support an active response, helping to quickly mobilize body systems. OT interacts with all of these, and may subsequently re-establish homeostasis and precondition the organism to deal with future stressors. However, the individual history of an organism, including epigenetic modifications of classical stress hormones such as VP, can moderate the effects of OT. Oxytocin’s effects also help to explain the important role of sociality in mammalian resilience and longevity. A hormetic perspective, focusing on a dynamic VP-OT system, offers new insights into emotional and physical disorders, especially those associated with the management of chronic stress, and helps us to understand the healing power of social behavior and perceived safety.

## 1. Purpose and Historical Background

In this narrative review, we hypothesize that a system involving interactions between oxytocin (OT) and vasopressin (VP) plays a critical role in understanding the physiological process termed “stress-response hormesis” [1]. VP and OT jointly encode environmental experience in cellular memory and create mechanisms that can be leveraged to optimize adaptation across many domains of human health. Although not previously recognized, many of the functions attributed to VP-OT interactions are similar to those described in hormesis.

### 1.1. Oxytocin and Vasopressin

Clinical research in the early 20th century recognized the capacity of extracts of the pituitary gland to speed up labor or facilitate milk ejection. OT was chemically identified as a polypeptide in the 1950s, an achievement recognized with a Nobel Prize [2]. However, as recently as the 1980s, research papers continued to describe OT as a “female” reproductive hormone with “no known function in males” [3]. The narrow characterization of OT as of importance only in females and only relevant to reproduction was incomplete and almost certainly slowed research on this remarkable molecule [4,5]. The pleiotropic functions of OT are only now being recognized [6,7].

OT extends beyond social and reproductive functions, including the regulation of food intake (reducing appetite via hypothalamic pathways), inducing muscle contraction (notably during parturition and lactation), and influencing thermoregulation by activating brown adipose tissue and mediating cold stress responses [8] OT has been more recently identified as an “anti-stress” hormone [9,10] with major consequences for health, resilience [8] and longevity [11]. OT has anti-inflammatory properties and is involved in regulating oxidative stress [12,13]. Furthermore, pathways dependent on OT facilitate prosocial behaviors, creating a “social solution to the stress of life” [14]. However, OT acts against the background of many other molecules, including VP, corticotropin-releasing hormone (CRH), urocortin (UCN), and molecules of the immune system. The temporal effects of OT vary in this context. Although both oxytocin-like and vasopressin-like peptides emerged from a shared ancestral gene early in vertebrate evolution, our current understanding of their functional specialization is based largely on mammalian and rodent models. While we use evolutionary terminology to highlight adaptive roles, we acknowledge that direct functional evidence from lower vertebrates and invertebrates remains limited.

In this review, we focus on the relationship of OT with a more ancient neuropeptide, vasopressin. VP and OT arose from a common ancestor [15]. These structurally similar peptides exhibit dynamic cross-reactivity with each other’s receptors [16]. Together VP and OT form a system with the capacity for adaptive and sequential changes, and with varying consequences across time. OT may be particularly relevant in the context of chronic stress, where it helps protect against endocrine dysregulation in conditions such as posttraumatic stress disorder (PTSD). This may in part be due to OT’s capacity to sequentially counter the actions of VP [4,17,18] and temper physiological overreactions to threat.

Under chronic versus acute conditions, the phenotype of these responses is different: VP is associated with quick responses including mobilization and anxiety, while OT responds more slowly, and may be associated with calming, immobilization and restoration. For example, the VP-OT system is especially important in social behavior and attachment [19]. In social mammals, the presence of safe others is of particular relevance during stress-management and “sociostasis” [20]. We suggest here that when better understood, VP- and OT-mediated interactions will offer novel strategies to harness the therapeutic capacity of these neuropeptides.

### 1.2. Time Matters

In mammals under chronic stress, VP and OT molecules have divergent, interactive and often sequential functions. In addition, the receptors for VP and OT, along with their subcellular signaling mechanisms, are critical for enabling dynamic functional interactions between these systems [21].

Across the lifespan, time also matters. There is abundant evidence that the VP-OT system is adaptively calibrated by early life experiences [22,23]. Context also is critical in regulating the VP-OT system [24]. The hormetic perspective described below also emphasizes the importance of time, development, feedback systems and context—all processes within which VP and OT have been shown to have dynamically adaptive functions.

## 2. Hormesis

Hormesis as a process has been used to describe biphasic or cyclic responses to challenge across species, ranging from single-cell organisms to vertebrates, including humans [25,26]. For convenience, we are focusing our scope here on mammals. Although hormetic functions are observed in a broad array of living organisms, we recognize the specific endocrine systems described here may not generalize beyond mammals. However, as initially described [1], cellular and subcellular processes that are shared across phyla may offer unifying mechanisms that explain hormesis across levels of biological complexity.

### 2.1. The Role of Peptides in the Hormetic Hypothesis

Both VP and OT are released by various challenging or stressful experiences. This pattern of simultaneous or sequential release could help to explain the poorly understood, biphasic, and sometimes beneficial consequences of “stress,” especially if the challenge is followed by a period of restoration. This process, sometimes known as “stress-response hormesis”, has well-documented consequences for health, restoration and longevity [1,27].

VP and OT have independently been shown to play major roles in the regulation of stress [6,28]. The current review focuses on the novel hypothesis that VP and OT also are critical to “stress-response hormesis”. Here we suggest that, at least in mammals, the benefits of hormesis may involve the sequential effects of VP followed by a compensatory release of OT. To our knowledge, this hypothesis has not been experimentally tested in the context of hormesis. We provide selected examples of hormesis in which interactions among OT and VP may help to explain beneficial adaptations following intense challenges, such as those observed in birth or exercise (Table 1 and Table 2). We further suggest that the investigation of the hormetic role of these peptides may identify previously unidentified adaptive functions for both VP and OT, and deepen our understanding of their coordinated activity as an integrated system.

### 2.2. Classical Definitions of Hormesis

Hormesis is a highly conserved, biphasic biological process characterized by stimulatory or beneficial effects following a challenge, and inhibitory or deleterious effects at high doses or during prolonged exposure. Initially conceptualized within toxicology [29,30], hormesis has since evolved into a unifying paradigm that intersects diverse fields such as aging biology, neuroscience, immunology, metabolism, and exercise physiology [31] (Table 2).

**Table 2 cimb-47-00632-t002:** Translational applications of CACH in human healthspan and disease modulation. This table illustrates the translational relevance of CACH across key human domains. It outlines how intermittent, tolerable stressors engage catabolic signaling, followed by anabolic restoration and remodeling. The net effect is enhanced systemic resilience, longevity, and performance, with implications for preventive and therapeutic strategies in aging, cardiometabolic health, and psychiatry.

Domain	Example Stressors/Interventions	Mechanisms Catabolic–Anabolic Cycling	Key Outcomes	Ref.
Aging & Geroscience	Fasting Heat/cold exposure Exercise (e.g., resistance or endurance)	Catabolic: Autophagy, mitophagy, SIRT1/FOXO Anabolic: Mitochondrial biogenesis, mTORC1, IGF-1 Cellular repair, proteostasis	Extended healthspan Reduced frailty Improved mitochondrial function	[32,33]
Cardiovascular Resilience	High-Intensity Interval Training (HIIT) Remote ischemic preconditioning Mild oxidative stress (e.g., ozone, hypoxia exposure)	Catabolic: ROS → Nrf2/HSPs, shear stress signaling Anabolic: Angiogenesis, endothelial nitric oxide production, anti-inflammatory cytokines	Improved vascular function Reduced endothelial senescence Autonomic balance	[32,34]
Neuropsychiatric Health	Controlled emotional challenge Novelty and cognitive engagement Therapeutic stress (e.g., exposure therapy)	Catabolic: Acute HPA-axis activation, monoaminergic shifts Anabolic: Synaptic remodeling, BDNF, OT/VP modulation, epigenetic plasticity	Enhanced stress resilience Reduced depression/anxiety Cognitive flexibility	[33,35,36]

Hormesis is characterized by adaptive benefits arising from exposure to transient stressors. This is a biphasic phenomenon, wherein stress induces cellular and systemic mechanisms that feed forward to enhance resilience, repair, and longevity. Hormesis has garnered increasing recognition as a unifying paradigm across multiple biological disciplines [29,37,38].

In contrast to traditional views that regard stress as inherently deleterious, the hormetic model proposes that organisms can be preconditioned by moderate stress exposures to mount more effective responses to future challenges, thereby decreasing disease vulnerability and promoting systemic robustness [39,40].

### 2.3. Ancient Peptides and Hormesis

We suggest here that at the core of the neuroendocrine orchestration of stress-related hormetic adaptation are VP and OT, and their primary receptors. VP is the more primitive neuropeptide, believed to have evolved over 400 million years ago from vasotocin. OT-like molecules existed for millennia [15]. However, in its current form, OT appeared in conjunction with the evolution of mammals over 200 million years ago [15,29]. In mammals, both VP and OT are primarily synthesized in the paraventricular and supraoptic nuclei of the hypothalamus, although VP and OT usually appear in different cells [15,41]. These peptides exert widespread effects through both central and peripheral pathways, and are deeply implicated in modulating stress responsivity, social behavior, emotional processing, immune regulation, and metabolic balance.

VP is classically associated with the initiation of acute stress responses, operating primarily through the activation of the HPA axis. VP facilitates a suite of immediate survival-oriented adaptations, including elevated arousal, enhanced vigilance, cardiovascular activation, fluid retention, and mnemonic reinforcement [42,43]. While these responses are adaptive under conditions of short-term stress, sustained or dysregulated VP signaling has been implicated in pathological outcomes such as generalized anxiety, major depression, metabolic syndrome, and social dysfunction [18,35,36]. This dualistic capacity positions VP as a dose- and context-sensitive regulator. The controlled activation of VP supports the hormetic enhancement of stress tolerance, but chronic overactivation can promote allostatic overload.

OT also may be released by challenges, but OT seems to work more slowly than VP [20]. The functions of OT include the facilitation of the recovery and restorative phases of reactions to stressful experiences [10]. OT can downregulate HPA axis activity, mitigate inflammation, enhance immune surveillance, and promote parasympathetic reactivation. In mammals, these mechanisms are essential for the resolution of acute stress and the establishment of physiological homeostasis. Beyond its central neuromodulatory effects, OT has demonstrated systemic benefits, including the maintenance of gut barrier integrity, the modulation of gut microbiota, and the regulation of glucose and lipid metabolism, positioning it as a key player in metabolic resilience and healthy aging [44,45,46]. Furthermore, OT contributes to cognitive flexibility, emotional regulation, and the formation of prosocial bonds. OT, in conjunction with VP, maximizes the benefits of positive social experiences [20]. Collectively, OT interactions with VP may serve as buffers against chronic stress and even psychopathology [47,48].

Emerging research highlights a reciprocal and dynamic interaction between VP and OT, whereby these neuropeptides mediate the shift from acute stress activation to adaptive recovery. VP is temporally aligned with immediate threat detection and mobilization, whereas OT operates during the subsequent recalibration phase, fostering synaptic plasticity, tissue repair, and behavioral regulation. This biphasic neuropeptidergic modulation exemplifies the core principles of hormesis, enabling organisms to respond with both rapid adaptation and long-term resilience [46,49].

### 2.4. Hormesis: Conceptual Foundations

#### 2.4.1. Oxidative Stress

At the cellular level, one of the most well-characterized manifestations of hormesis is the oxidative stress response. Subtoxic exposure to reactive oxygen species (ROS) can initiate adaptive subcellular signaling cascades that upregulate antioxidant defenses [e.g., nuclear factor erythroid 2-related factor 2 (Nrf2) activation], enhance DNA repair, and promote autophagy and proteostasis, thereby increasing cellular longevity and functional resilience [32,34,50]. This concept extends into metabolic hormesis, as observed in caloric restriction, intermittent fasting, and other forms of nutrient stress, which activate energy-sensing pathways such as AMP-activated protein kinase (AMPK), Sirtuin 1 (SIRT1), and Peroxisome proliferator-activated receptor gamma coactivator 1-alpha (PGC-1α). These in turn induce mitochondrial biogenesis, improve oxidative phosphorylation efficiency, and reduce systemic inflammation, which are hallmarks of delayed aging and metabolic flexibility [33,51,52,53,54] (Table 2).

#### 2.4.2. HPA Axis Hormones and Hormesis

An essential upstream regulator of stress-induced hormesis is the CRH system, which coordinates central and peripheral responses to homeostatic perturbation. CRH, a hypothalamic neuropeptide released from parvocellular neurons in the paraventricular nucleus (PVN), plays a critical role in orchestrating the initial activation of the HPA axis, thereby facilitating energy mobilization, vigilance, and behavioral arousal in response to stressors. This activation represents the early catabolic phase of the hormetic cycle, promoting systemic readiness and cellular defense through glucocorticoid secretion and sympathetic stimulation [55]. Recent advances expanded the CRH paradigm to include its peripheral and centrally acting paralogs, the urocortins (UCN1, UCN2, and UCN3), which engage both CRH receptor type 1 (CRHR1) and type 2 (CRHR2) with divergent physiological outcomes.

CRHR1 activation by CRH and UCN1 is typically associated with acute stress mobilization and anxiety-like behavior, mirroring VP-mediated HPA axis activation and catabolic mobilization [56,57]. Conversely, CRHR2, primarily activated by UCN2 and UCN3, has been shown to promote stress recovery, cardiovascular stability, and anxiolytic effects by functioning as a molecular mediator of the anabolic, restorative phase of the hormetic response [56]. These receptor-specific dynamics recapitulate the broader catabolic–anabolic oscillation seen across hormetic systems and align with the neuropeptide dichotomy observed in VP and OT signaling.

Moreover, urocortins (UCNs) act at both central and peripheral sites, including the heart, gastrointestinal tract, and immune system, where they modulate inflammation, oxidative stress, and metabolic regulation, features consistent with broader hormetic adaptation [58,59]. UCN2 and UCN3, in particular, have been implicated in enhancing cardiovascular resilience and attenuating stress-induced autonomic dysfunction, thereby serving as potent endogenous hormetins in their own right. Their biphasic effects are tightly regulated by tissue-specific receptor expression and feedback sensitivity, thus rendering them highly responsive to stress intensity, duration, and context.

Importantly, the CRH-UCN system also interfaces with VP and OT pathways to fine-tune behavioral and physiological outcomes. CRH and VP act synergistically at the pituitary to enhance ACTH release, amplifying glucocorticoid output during acute stress, while urocortin-driven CRHR2 activation counterbalances this effect by dampening HPA axis activity and promoting parasympathetic recovery, mechanistically resembling OT-mediated stress resolution [57,60]. These intersecting networks underscore a hierarchically organized stress-adaptation system, wherein CRH family peptides initiate and regulate the transition between the mobilization and restitution phases of hormesis.

The CRH-UCN axis represents a critical neuroendocrine interface between environmental challenges and systemic adaptation. Its inclusion in the hormetic model enriches our understanding of how organisms maintain stability, growth and restoration through change, using precisely timed, receptor-specific neuropeptide signaling to convert stress exposure into long-term physiological gains. Future research into selective CRHR2 agonists or UCN mimetics may offer novel therapeutic avenues for enhancing resilience, particularly in individuals with dysregulated HPA axis dynamics or impaired stress recovery.

Hormesis has particular relevance in the context of neuroprotection and cognitive resilience, where mild cognitive challenges and psychological stressors can stimulate neurogenesis, synaptic remodeling, and increased resistance to neurodegenerative processes. These forms of behavioral hormesis engage molecular mechanisms such as brain-derived neurotrophic factor (BDNF) upregulation, mitochondrial adaptation, and glutamatergic signaling balance [61,62]. Importantly, stress resilience is not merely a function of exposure magnitude, but of precise timing, context, and neuroendocrine feedback, features central to the hormetic framework.

Within this adaptive schema, neuropeptides such as VP and OT [10] have emerged as critical regulators of systemic stress responses. VP predominantly facilitates catabolic mobilization, acting via HPA axis activation, sympathetic upregulation, and vasoconstrictive pathways. In contrast, OT is associated with anabolic recovery, vagal activation, and anti-inflammatory signaling [63,64]. Together, these neuropeptides mediate a bi-phasic oscillation between challenge and recovery, positioning the components of the VP-OT system as hormonal drivers of stress-response hormesis [65].

Integrating VP and OT into the broader framework of hormesis reveals a neuroendocrine model of adaptation, in which the strategic alternation between mobilization and restitution shapes both physiological and behavioral outcomes. This paradigm has wide-ranging implications, from cardiometabolic adaptation and neuropsychiatric resilience to interventions in aging and neurodegenerative disease, particularly those that exploit cyclic stress exposures, such as intermittent fasting, cold/heat stress, high-intensity interval training, and controlled psychosocial engagement [47,66,67].

Hormesis is also closely linked to stress resilience and adaptation, particularly in the nervous system. Exposure to mild psychological stress can enhance cognitive function and emotional resilience, a phenomenon described as behavioral hormesis [46]. In this context, neuropeptides such as VP and OT play a pivotal role in stress adaptation, helping to regulate the balance between the challenge phase (VP-driven) and the recovery phase (OT-driven) [47] Figure 1.

By incorporating VP and OT into the hormetic framework, a novel perspective emerges in which these neuropeptides contribute to stress-response hormesis by dynamically modulating physiological and behavioral adaptation processes Figure 1. The dynamic involvement of VP and OT as dual-phase modulators expands this framework, offering a translational lens through which to reframe therapeutic strategies in stress-related pathology, age-associated decline, and chronic disease prevention.

## 3. Catabolic–Anabolic Cycling Hormesis Model: A Dynamic Framework for Adaptive Health

The CACH model represents an emerging conceptual framework that extends classical hormesis into a temporal and phase-dependent paradigm, wherein biological systems alternate between stress-induced catabolic activation and compensatory anabolic recovery. Unlike linear models of stress adaptation, CACH emphasizes the rhythmic and cyclical nature of organismal resilience, anchoring the health-promoting benefits of transient perturbations in their capacity to trigger restorative, regenerative, and homeostatic processes.

### 3.1. Theoretical Foundations and Examples

At its core, CACH integrates the well-established biphasic dose–response curve of hormesis with biological oscillation theory, proposing that optimal adaptation requires not just the right magnitude of stimulus, but also precise timing, duration, and recovery dynamics [66,68]. The catabolic phase initiates cellular stress responses, e.g., via ROS signaling, AMP/ATP ratio sensing, or glucocorticoid mobilization, activating defense pathways such as autophagy, mitochondrial uncoupling, and immune priming. This is followed by an anabolic phase, where growth factors, anti-inflammatory mediators, and mitochondrial biogenesis coordinate structural repair and functional recalibration. The coordination between these opposing yet complementary forces allows for efficient adaptation to environmental stressors without tipping into maladaptation or exhaustion Table 3.

### 3.2. CRH, Urocortins, and the Neuroendocrine Modulation of CACH

CRH and UCNs play a foundational role in orchestrating the initiation, regulation, and resolution of stress responses within the CACH framework. CRH acts as the principal neuropeptide initiating the catabolic phase of stress adaptation. It stimulates ACTH release, thereby activating the HPA axis and promoting systemic glucocorticoid release [55]. This axis supports energy mobilization, immune vigilance, and increased arousal, all of which align with early-phase hormetic stress activation.

However, the CRH system does not act in isolation. Its paralogs, the UCNs, modulate stress responsiveness in a receptor- and phase-specific manner, aligning tightly with the cyclical logic of CACH. UCN1 binds to both CRHR1 and CRHR2 and plays a role in initiating the early stress response, while UCN2 and UCN3 exhibit preferential binding to CRHR2, a receptor system associated with recovery, cardiovascular stability, and anxiolytic signaling [56,70]. This receptor divergence reflects a mechanism for transitioning from catabolic stress mobilization to an anabolic resolution, which is a hallmark of CACH’s temporal organization.

CRHR2 activation, in particular, is implicated in parasympathetic rebound, cardiac repair, immune rebalancing, and behavioral recovery, functions similar to OT’s role in the anabolic phase [70]. UCNs, therefore, act as endogenous hormonal switches, complementing VP and OT by refining the amplitude, duration, and systemic scope of stress-phase transitions. These peptides are expressed not only in the brain but also in peripheral tissues such as the heart, gut, and vasculature, where they modulate inflammatory tone, vascular remodeling, and redox homeostasis [58,59].

Moreover, CRH and VP synergistically potentiate ACTH release during acute stress, amplifying the mobilization of energy substrates and behavioral arousal [43]. This cooperation ensures that the catabolic phase is robustly activated during high-salience stress exposures. In contrast, OT and CRHR2 agonism (via UCN2/3) serve to terminate the stress response, engaging vagal pathways and promoting anti-inflammatory, neurotrophic, and tissue-regenerative processes [46]. The opposing yet coordinated actions of these peptide systems provide molecular substantiation for the phase-specific dynamics of the CACH model.

Taken together, the CRH–UCNs axis functions as an upstream timing mechanism, activating early catabolic defenses and coordinating their resolution through receptor-specific feedback loops. When viewed through the lens of the CACH framework, this neuroendocrine system reinforces the cyclical nature of adaptive physiology, mediating transitions between stress engagement and reparative recalibration. Integrating CRH and UCNs alongside VP and OT enhances our understanding of the molecular choreography underlying resilience, and provides a more nuanced platform for designing phase-targeted interventions across diverse clinical domains.

## 4. Vasopressin and Oxytocin as Endocrine Drivers of CACH

In addition to CRH and UCNs, VP and OT play an important role in orchestrating the catabolic–anabolic axis of stress adaptation. VP is predominantly active during the catabolic stress phase, enhancing sympathetic tone, vasoconstriction, and HPA axis activity. In contrast, OT becomes prominent during the anabolic recovery phase, supporting vagal activation, anti-inflammatory signaling, endothelial repair, and mitochondrial stabilization [63,65]. This bidirectional neuroendocrine rhythm aligns with autonomic cycling (sympathetic vs. parasympathetic), and can be entrained or modulated via behavioral, environmental, or pharmacologic stimuli.

## 5. Stress

Stress constitutes a multidimensional construct encompassing psychological, physiological, and environmental challenges to an organism’s homeostasis. Within the framework of hormesis, stress functions not solely as a pathological insult, but as a potentially salutogenic stimulus capable of inducing adaptive biological responses when administered within an optimal dose–response window. The biphasic nature of the hormetic curve, with beneficial effects at low-to-moderate intensities and deleterious consequences at high exposures, underscores the critical importance of allostatic regulation, neuroendocrine coordination, and intracellular signaling in mediating organismal resilience [71,72].

### 5.1. Neuroendocrine Architecture of the Stress Response

The stress response is orchestrated by the neuroendocrine system, prominently featuring the HPA axis and what is sometimes described as the sympatho-adreno-medullary (SAM) system. Upon perception of a stressor, the PVN of the hypothalamus is activated, leading to the co-secretion of CRH and VP that act synergistically on corticotroph cells in the anterior pituitary to promote ACTH synthesis and release, culminating in the secretion from the adrenal cortex of glucocorticoids, primarily cortisol in humans and corticosterone in rodents [55]. Concurrently, the stress-induced activation of the locus coeruleus–norepinephrine (LC-NE) system and the adrenal medulla facilitates a rapid catecholaminergic response, driving cardiovascular, metabolic, and attentional adaptations necessary for immediate survival [73].

### 5.2. Vasopressin and Oxytocin in the Central Stress Network

Both VP and OT, synthesized predominantly by the magnocellular and parvocellular neurons of the PVN and supraoptic nucleus (SON), modulate the stress response through complex feedback and feedforward loops. VP, acting through V1b receptors in the anterior pituitary, enhances CRH-induced ACTH secretion, particularly during chronic or repeated stress, thereby sustaining HPA axis activity and potentially contributing to hypercortisolemia [46,55].

OT, acting through its G-protein coupled receptor expressed in limbic and hypothalamic circuits, exerts anxiolytic and stress-attenuating effects, partly by dampening amygdala reactivity and modulating central autonomic output [74]. OT’s regulatory effects on the autonomic nervous system can also contribute to long-term social bonding and adaptive recovery following stress [6,75,76,77]. VP and OT not only influence endocrine outputs but also serve as neuromodulators, dynamically regulating synaptic plasticity, emotional processing, and social cognition. Their sometimes antagonistic but functionally complementary effects reveals a calibration of social and environmental adaptability, with VP facilitating vigilance and defensive behaviors, and OT promoting affiliative bonding and the social buffering of stress [35].

## 6. Hormetic Stress: Cellular Adaptation and Systems Resilience

The concept of hormetic stress challenges the traditional dichotomy of stress as either “good” or “bad,” instead proposing a dose-dependent continuum wherein subtoxic exposures to stressors (e.g., heat shock, hypoxia, oxidative stress, caloric restriction, physical exercise, or psychosocial novelty) activate cellular defense pathways. These include the Nrf2 antioxidant system, HSPs, AMPK, sirtuins (e.g., SIRT1), and autophagic machinery, which collectively confer cytoprotection, neuroplasticity, and metabolic efficiency [71,78].

Stress-induced neuropeptides play integral roles in these hormetic adaptations. For instance, VP has been implicated in energy homeostasis and osmoregulation, linking neuroendocrine stress responses to mitochondrial efficiency and metabolic hormesis [79]. OT, conversely, has been shown to reduce ROS production and inflammation while enhancing parasympathetic tone and social reinforcement learning, positioning it as a neuromodulator of stress resilience and organismal recovery [6,80]. It was shown that OT evolved as a molecular safeguard against oxidative stress, acting at both systemic and mitochondrial levels to preserve cellular resilience [81].

### 6.1. Chronic Stress, Allostatic Load, and Pathophysiological Transition

When stress becomes chronic, cumulative, or dysregulated, the physiological benefits of hormesis are replaced by allostatic load, the wear and tear resulting from the prolonged activation of adaptive systems [82]. Under such conditions, VP expression is upregulated, often independently of CRH, sustaining elevated HPA axis activity and glucocorticoid output, which can lead to hippocampal atrophy, insulin resistance, immunosuppression, and mood dysregulation [60,83]. Simultaneously, chronic stress blunts the oxytocinergic system, impairing social functioning and increasing vulnerability to affective disorders such as depression, anxiety, and PTSD [84]. These impairments may result from the downregulation of oxytocinergic tone and social buffering mechanisms, which have been described as critical mediators of resilience under adversity [48].

Notably, the differential epigenetic regulation of VP and OT genes in response to early-life adversity or trauma further compounds stress susceptibility across the lifespan, supporting a developmental programming model of stress neurobiology [84]. This underscores the clinical relevance of these neuropeptides not only in acute stress modulation, but also in long-term neuropsychiatric trajectories. Although oxytocin supports healthy aging processes, its concentration generally decreases with age due to its association with reproductive function.

### 6.2. Integration into the Hormetic Framework: Oxytocin and Vasopressin as Bidirectional Modulators of Allostatic Flexibility

Integrating VP and OT into a hormetic model of stress physiology requires recognizing their reciprocal roles in regulating allostatic flexibility, the organism’s capacity to mount, modulate, and recover from neuroendocrine and behavioral responses across varying intensities of environmental demands. Hormesis, characterized by a biphasic dose–response pattern wherein moderate stress induces adaptive resilience while excessive stress elicits pathology, hinges on temporally coordinated activity across central and peripheral signaling networks [85].

VP contributes primarily to the reactive phase of allostasis by potentiating HPA axis activity. Via V1b receptor activation, VP synergizes with CRH to promote ACTH secretion and downstream glucocorticoid release, thus enhancing metabolic readiness and cardiovascular tone [86]. However, chronic VP signaling can be associated with allostatic overload, contributing to neuroendocrine rigidity, glucocorticoid resistance, and behavioral inflexibility, hallmarks of maladaptive stress reactivity seen in affective and anxiety disorders [46].

Conversely, OT is preferentially engaged during the resolution and recovery phases of the stress cycle. Central OT receptor (OTR) activation in limbic and hypothalamic circuits, such as the amygdala and ventral hippocampus, attenuates CRH neuron activity, dampens amygdalar hyperexcitability, and re-establishes parasympathetic dominance [46]. OT also facilitates social buffering, reduces neuroinflammation, and enhances hippocampal plasticity and neurogenesis, contributing to the restoration of homeodynamic equilibrium [46,87].

Together, VP and OT define a bidirectional regulatory axis that orchestrates both the mobilization and resolution of stress responses. While VP facilitates immediate systemic robustness during threat exposure, OT supports plasticity, affiliative behavior, and physiological recalibration. The dysregulation of this axis, particularly VP overactivation coupled with insufficient OT tone, has been implicated in the pathophysiology of post-traumatic stress disorder (PTSD), depression, and social dysfunctions [46,88]. This bidirectional modulation supports the conceptualization of OT and VP as evolutionary complements [48], balancing threat detection with social healing and neuroendocrine flexibility [89].

Therefore, the balance between VP and OT signaling constitutes a central mechanism of neuroendocrine hormesis, modulating not only acute stress reactivity but also long-term resilience and adaptive capacity. Recent findings suggest that therapeutic interventions targeting this axis, such as intranasal OT, VP antagonists, or neuromodulation strategies, may enhance psychological recovery, reduce stress-related morbidity, and mitigate cognitive decline in aging populations [90]. However, individual differences in this system are common, and in part may reflect the cumulative properties of hormesis across the lifespan.

### 6.3. Vasopressin and Oxytocin as Hormetic Modulators

Beyond VP and OT’s classical roles in osmoregulation and parturition, they function as modulators of the stress response, exhibiting properties consistent with hormetic agents. We have previously discussed the integrative roles of OT and VP in social attachment and their capacity to modulate stress responses [91]. Their effects are highly context-dependent, influenced by factors such as dosage, exposure duration, receptor subtype activation, and the specific neural circuits involved. We have shown that the behavioral effects of OT and VP vary depending on the perceived emotional context and individual history, reflecting their complex roles in social behavior [89]. At optimal levels, VP and OT can promote adaptive recovery and resilience; however, dysregulation, whether through excessive or insufficient signaling, may lead to maladaptive outcomes.

VP has been implicated in the modulation of social behaviors and stress responses. For example, our prior research in prairie voles indicated that VP plays a crucial role in social bonding and attachment, with its effects being modulated by early social experiences [48]. Studies indicate that VP can influence aggression and social bonding, with effects varying based on administration route and receptor interaction. In another example, in socially isolated mice, VP administration dose-dependently inhibited heightened aggression, suggesting a nuanced role in social behavior modulation [57]. Conversely, chronic elevated VP levels have been associated with increased stress reactivity and anxiety-like behaviors, highlighting the importance of balanced VP signaling.

OT facilitates social bonding and reduces stress. Its administration has been shown to decrease nonsocial risk-based decision-making, indicating a potential role in enhancing prosocial behaviors [92]. OT’s actions are influenced by emotional context and individual experiences, contributing to its role in social attachment and stress regulation [93]. However, the effects of OT are complex and can be influenced by factors such as sex, context, and individual differences. For example, OT has been found to exert sex-specific effects on social behaviors, with variations observed in aggression and pair bonding [94]. Additionally, excessive OT stimulation may lead to receptor desensitization or disrupt synaptic homeostasis, underscoring the importance of dosage context and individual histories in its therapeutic application.

The cellular interplay between VP and OT systems adds yet another layer of complexity. Both peptides can bind to each other’s receptors, leading to potential crosstalk and varied behavioral outcomes [89]. The interactions between OT and VP are crucial for understanding their roles in social behaviors, as their effects can be paradoxical and context-dependent. The activation of OTRs has been associated with both pro-social and, in certain contexts, pro-aggressive behaviors, potentially mediated by endogenous OT. In contrast, the administration of synthetic OT or VP has been reported to reduce aggression, likely through the activation of vasopressin V1a receptors [95]. This apparent bidirectional modulation underscores the importance of context, dosage, and receptor distribution, suggesting that the behavioral effects of OT and VP depend on a finely tuned balance between their signaling pathways.

In therapeutic contexts, understanding the hormetic properties of VP and OT is essential. For example, the intranasal administration of OT has been explored as a potential treatment for social deficits in neuropsychiatric disorders. The potential for targeting OT and VP pathways for therapeutic interventions in social deficits is associated with neuropsychiatric disorders, highlighting the importance of individualized approaches [48]. Outcomes of interventions, such as intranasal OT, are often variable, and factors such as dosage, individual sensitivity, and the specific behavioral context play significant roles in determining efficacy [96]. Similarly, VP receptor antagonists are being investigated for their potential to mitigate stress-related disorders, but careful titration is necessary to avoid disrupting essential physiological functions. In addition, because VP and its receptors appear to be calibrated in early life by adversity or nurturing, the effects of manipulations of the VP system may appear paradoxical. These unexpected or undesired outcomes could possibly be explained by cross-talk between OT, VP and their receptors, as well as differential sensitivity of these receptors.

Taken together, the current literature on VP and OT suggests that these are integral components of the neuroendocrine system that exhibit hormetic properties, modulating stress responses and social behaviors in a context-dependent manner. Their dualistic nature, as facilitators of both adaptation and potential maladaptation, highlights the importance of precise regulation and the need for nuanced therapeutic approaches that consider individual variability and environmental context.

### 6.4. Vasopressin’s Role in Neuroendocrine Plasticity

VP is increasingly recognized not only as a classical endocrine hormone involved in fluid balance and vasoconstriction, but also as a central neuromodulator with profound effects on neuroendocrine plasticity. Its ability to reshape neural circuits, tune stress reactivity, and mediate behavioral outcomes reflects its integration into the dynamic, hormetic architecture of the stress system. Within this framework, VP supports both immediate adaptation to challenges and long-term recalibration of the HPA axis and limbic processing through activity-dependent, epigenetic, and receptor-specific mechanisms.

#### 6.4.1. Structural and Functional Plasticity of Vasopressin Neurons

VP is synthesized primarily by magnocellular neurons in the PVN and SON of the hypothalamus, as well as parvocellular neurosecretory cells that co-release CRH. These neurons exhibit notable structural plasticity in response to environmental stimuli such as osmotic imbalance or stress exposure. For example, dendritic hypertrophy, increased synaptic input, and glial remodeling have been observed in VP neurons during dehydration and chronic stress, enhancing neurosecretory efficiency [97]. This plasticity supports the concept of neuroendocrine metaplasticity, the long-term modulation of hormone release dynamics based on prior experiences [98]. At the synaptic level, VP neuron excitability is modulated by glutamatergic, GABAergic, nitric oxide, and astrocytic signaling, each of which contributes to flexible hormonal output [99]. This responsiveness allows VP networks to act as finely tuned transducers of internal and external signals, consistent with hormetic regulation.

#### 6.4.2. Epigenetic Regulation and Early-Life Programming

VP gene expression is subject to epigenetic modifications, which can program long-lasting changes in stress physiology. As one example, early-life adversity, such as maternal separation, leads to the hypomethylation of VP promoter/enhancer regions in the PVN, resulting in the persistent upregulation of VP and heightened HPA axis reactivity in adulthood. These molecular changes are part of an adaptive developmental plasticity system that may confer survival advantages in stressful environments but increase vulnerability to anxiety or depressive disorders if environmental predictability is low [84]. VP-sensitive circuits in the bed nucleus of the stria terminalis (BNST), central amygdala, and lateral septum also undergo experience-dependent changes in connectivity and receptor expression, contributing to the modulation of aggression, fear, and social memory [100]. Developmental studies in prairie voles have provided compelling evidence that early-life manipulations of VP pathways, including neonatal handling or pharmacological exposure, can program long-lasting changes in VP receptor expression and aggression, reflecting a form of early-life neuroendocrine imprinting that parallels epigenetic modulation [101].

#### 6.4.3. Receptor-Specific Signaling and Behavioral Adaptation

VP acts on three G-protein-coupled receptors—V1aR, V1bR, and V2R—of which V1aR and V1bR are predominantly expressed in the brain. The activation of V1aR in the amygdala and septum facilitates territorial aggression, social memory, and pair bonding, while V1bR activation in the anterior pituitary stimulates ACTH release and augments HPA axis tone [35,102]. Recent optogenetic and chemogenetic studies have shown that the temporal patterning of VP neuron activity in the extended amygdala can bidirectionally control anxiety-like behavior, suggesting that VP’s behavioral impact is tightly linked to network dynamics and circuit state [95,103]. Consistent with these findings, it was shown in prairie voles that developmental VP exposure can lead to increased adult aggression, highlighting the enduring impact of receptor-mediated signaling during critical periods of brain plasticity [104].

#### 6.4.4. Integration of Vasopressin into Hormetic Stress Networks

VP’s role in hormetic adaptation lies in its capacity to prime the stress response system under moderate stress while enabling structural remodeling for future responsiveness. Subthreshold VP release can enhance neurotrophic signaling, glucose metabolism, and attentional resources, thereby optimizing brain function under load [105]. However, when VP signaling becomes chronic or dysregulated, particularly in the absence of counter-regulatory oxytocinergic tone, it may contribute to allostatic overload, anxiety disorders, or metabolic dysregulation [46,86]. These outcomes support the general hypothesis that OT and VP act as neurochemical pivots between adaptive cooperation and defensive reactivity, depending on environmental cues and internal states [89]. Thus, VP exemplifies a bidirectional neuroendocrine effector, capable of mediating resilience or pathology depending on dosage, timing, receptor specificity, and environmental context. Its capacity to regulate plasticity at molecular, cellular, and network levels situates it as a central modulator within the hormetic stress architecture.

### 6.5. Oxytocin’s Behavioral and Cellular Roles in Hormetic Regulation

OT is not simply as a “prosocial hormone” but also serves as a context-sensitive regulator of homeostasis. The actions of OT align closely with several benefits attributed to hormesis, and especially the adaptive calibration of biological systems in response to mild or moderate challenges. We specifically hypothesize here that OT may play a pivotal role in what has been called stress-response hormesis.

#### 6.5.1. Context-Dependent Modulation of Stress and Behavior

OT’s effects on stress and social behavior are profoundly context-dependent, shaped by internal state, early-life experiences, social environment, and OTR expression patterns [106]. Under moderate, predictable psychosocial stress, endogenous OT release facilitates social engagement, affiliative behavior, and stress attenuation. These effects are supported by evidence showing OT-induced amygdala deactivation and enhanced functional connectivity between the amygdala and prefrontal cortical regions, promoting top-down emotional regulation [69,87].

Such outcomes are consistent with a hormetic interpretation—low to moderate adversity stimulates adaptive neurobehavioral responses that increase the organism’s resilience to future stressors [107]. However, in contexts involving unpredictable social threat or early-life adversity, OT may actually enhance social vigilance, ingroup bias, or even anxiogenesis. This outcome has been interpreted to indicate that OT’s principal role is to amplify the salience of social stimuli rather than universally promote prosociality [108]. This dual action reflects OT’s function as a salience modulator, calibrating behavioral responses in a dose- and context-sensitive manner that is emblematic of hormetic signaling. Although oxytocin has been associated with stress attenuation and recovery, it is not inherently a calming hormone. Instead, OT release often follows stimulating challenges such as endurance exercise, parturition, cold exposure, fasting, or thermoregulatory stress, reflecting its role as a dynamic modulator of adaptive responses rather than a simple stress buffer.

#### 6.5.2. Cellular Mechanisms of Oxytocin-Mediated Hormesis

At the cellular level, OT promotes neuroprotection, synaptic plasticity, and oxidative resilience, hallmarks of cellular hormesis. OTRs are broadly expressed across the central nervous system, immune tissues, and cardiovascular system, enabling OT to exert cytoprotective and anti-inflammatory actions [46]. OT may have evolved as a molecular mechanism for regulating oxidative stress and mitochondrial health, supporting its role in cellular resilience and inflammation control [81]. In neurons, OT enhances mitochondrial efficiency, modulates intracellular calcium dynamics, and reduces oxidative stress through the upregulation of antioxidant enzymes such as superoxide dismutase (SOD) and glutathione peroxidase [109].

OT also facilitates hippocampal neurogenesis and synaptic remodeling, particularly under conditions of intermittent stress exposure, via pathways that overlap with classic hormetic mediators such as BDNF, MAPK/ERK, and PI3K/Akt signaling cascades [110]. These actions not only support cognitive flexibility and emotional regulation, but also protect against neurodegenerative processes. Furthermore, OT suppresses NF-κB activation, reducing pro-inflammatory gene expression and promoting immune tolerance under low-grade immune challenges [110].

#### 6.5.3. Social Hormesis: Affiliative Behavior as Adaptive Stressor

Recent theories have proposed the notion of social hormesis, whereby moderate social demands, such as cooperation, novelty, or conflict resolution, act as adaptive stressors that strengthen emotional and cognitive resilience [48,111]. OT plays a central role in this process: its release in response to social touch, eye contact, and shared goal pursuit not only enhances social bonding, but also initiates stress-buffering physiological cascades [109]. These mechanisms enable the organism to remain engaged in complex social environments without succumbing to chronic stress. As our previous work [48] argues, OT-mediated affiliative behaviors are evolutionarily conserved mechanisms of biobehavioral regulation, wherein moderate social challenges enhance cooperative behavior, stress tolerance, and long-term adaptive capacity. Oxytocin’s evolutionary role includes shifting behavioral priorities from food intake toward reproductive behaviors and mate-seeking, which helps explain its diverse behavioral effects.

Importantly, low-dose exogenous OT administration in clinical and experimental contexts has been shown to reduce cortisol levels, increase parasympathetic tone, and enhance threat appraisal accuracy, which are outcomes that reflect hormetic optimization of the stress axis [96]. However, sustained or high-dose administration may lead to OTR desensitization or network dysregulation, again underscoring the hormetic principle that “more is not necessarily better”, and that timing, context, and dose are critical determinants of function.

For instance, music-based social interactions such as group singing or rhythmic coordination have been shown to stimulate oxytocinergic activity and enhance prosocial bonding [112], supporting the idea of “social hormesis” through shared sensory and emotional experience.

OT exemplifies a bi-directional hormetic modulator, operating across behavioral and cellular domains to optimize organismal function under moderate challenge. Its ability to both buffer stress and amplify social salience, depending on the environmental and internal context, makes it a precision regulator of adaptive capacity, with broad implications for neuropsychiatric resilience, aging, and immuno-metabolic health.

## 7. Therapeutical Implications and Translational Opportunities

Framing OT and VP within a hormetic framework repositions these peptides not as binary stress mediators but as precision regulators of stress-response plasticity. Their therapeutic utility lies in the capacity to modulate neuroendocrine tone, behavior, and systemic function in a dose-, context-, and state-dependent manner.

### 7.1. Precision Neuropeptide Modulation

OT-based therapeutics, particularly intranasal administration, have demonstrated promise in treating autism spectrum disorder (ASD), schizophrenia, borderline personality disorder, and social anxiety disorder, via modulation of social salience and affective processing [90]. However, clinical heterogeneity remains a major challenge. Efficacy may depend on OTR gene polymorphisms, attachment style, and early-life adversity, suggesting the need for personalized interventions. Trials incorporating functional neuroimaging, genetic markers, and hormetic biomarkers (e.g., stress reactivity profiles) could help stratify individuals based on their position on the hormetic response curve [87,96]. This aligns with the proposal that individual responsiveness to OT may depend on the developmental calibration of receptor systems, which could serve as biomarkers for tailored intervention strategies [48].

Similarly, V1bR antagonists have been investigated for the treatment of major depressive disorder (MDD), generalized anxiety disorder, and alcohol dependence, owing to their ability to attenuate hyperactive HPA axis signaling [113]. In contrast, perhaps selective VP agonists could be advantageous under conditions of cognitive fatigue or circadian disruption, as VP promotes wakefulness, working memory, and social vigilance, particularly when administered in short-term, hormetic dosing paradigms [46,105].

### 7.2. Hormetic Interventions and Lifestyle Medicine

Integrating neuropeptidergic modulation with lifestyle-based hormetic stressors offers synergistic translational potential. Lifestyle interventions, such as exercise, intermittent fasting, cold exposure, and cognitive novelty, are now recognized as controlled stressors that engage hormetic pathways via mitochondrial, anti-inflammatory, and neuroplastic mechanisms [61]. The following are examples.

Physical activity increases endogenous OT release, enhances hippocampal neurogenesis, and upregulates BDNF, suggesting a synergistic mechanism that could be optimized in rehabilitation protocols for PTSD, MDD, and neurodegenerative conditions [87].

Mindfulness-based interventions, attachment therapy, and breathwork have demonstrated effects on OT and VP regulation, recalibrating HPA axis tone and reducing allostatic load, especially in populations with trauma exposure or early-life stress [114]. Such interventions may work in part by engaging conserved oxytocinergic circuits that evolved to buffer oxidative and psychosocial stress, providing a physiological rationale for combining behavioral and biochemical strategies [81].

## 8. CRH, Urocortins, and the Hormetic Stress Response—Setting the Stage for Oxytocin and Vasopressin

While OT and VP are central to social, emotional, and metabolic adaptations to stress, their functional integration within a hormetic model is best understood against the broader landscape of the HPA axis. At the heart of this system lies the peptides, CRH and UCNs, which act as frontline mediators of stress and adaptive recalibration.

### 8.1. CRH as a Hormetic Initiator

Hormesis, characterized by a biphasic dose response, describes how exposure to mild or moderate stress can activate adaptive processes, whereas excessive stress becomes detrimental. The HPA axis, regulated primarily by CRH, is a central mediator of these adaptive and maladaptive outcomes. Within the broader context of hormesis, CRH is not simply a stress-activating molecule, it is a conditioning signal, shaping both immediate responses and long-term adaptive plasticity. However, within the framework of hormesis, where low-level stress enhances long-term resilience, CRH’s role becomes more nuanced. It is not merely a trigger for downstream activation; CRH also serves as a neuroendocrine “primer” that sets the stage for both acute adaptation and long-term stress conditioning. In other words, CRH operates as a molecular threshold detector, helping the brain and body assess the severity of a challenge and calibrate the physiological response accordingly [107].

#### 8.1.1. Catabolic Phase Activation: Metabolic and Circadian Modulation

CRH, released from the hypothalamic PVN, drives ACTH release from the pituitary and subsequent cortisol/corticosterone secretion. These hormones mobilize metabolic fuels, glucose, fatty acids, and amino acids, supporting energy-intensive processes during stress [115]. The acute elevation of cortisol facilitates gluconeogenesis, lipolysis, and amino acid catabolism, while transient insulin resistance conserves glucose for vital organs [116]. Fasting and exercise-induced CRH release also promotes autophagy, growth hormone secretion, and BDNF expression for neuroplastic benefits [61].

CRH regulates brown adipose tissue (BAT) thermogenesis via sympathetic drive and Uncoupling protein 1 activation, enhancing energy expenditure and cold-induced resilience [117,118]. This is complemented by thyroid and cortisol synergy, sex differences in estrogen-mediated BAT recruitment, and therapeutic implications for metabolic disease [119]. CRH also interacts with the suprachiasmatic nucleus to align cortisol rhythms with circadian timing, modulating glucocorticoid receptor sensitivity [120]. Circadian disruption via sleep loss or light pollution impairs this synchrony, increasing disease risk [121]. Paradoxically and in contrast, mild circadian challenge, like intermittent fasting, may improve resilience through CRH-mediated entrainment [122].

#### 8.1.2. Epigenetic Reprogramming and Intergenerational Stress Signatures

CRH signaling induces epigenetic remodeling, DNA methylation, histone modification, and microRNA regulation, which persistently alters stress responsivity [123]. These modifications can be inherited, shaping behavioral traits and HPA tone across generations. Crucially, these epigenetic effects are reversible. Exercise, social support, and mindfulness interventions restore adaptive CRH regulation [124]. Our studies and others showed that OT and VP also interact with CRH-modulated epigenetic pathways, influencing bonding, emotional resilience, and social cognition [48,125,126].

### 8.2. CRH as a Conditioning Signal

#### 8.2.1. Immediate Response Calibration

In the short term, CRH helps orchestrate a precisely scaled response. Rather than triggering an all-or-nothing alarm, CRH levels rise in proportion to the intensity and duration of the threat, thereby controlling the magnitude of cortisol release, sympathetic tone, arousal levels, and immune modulation. This titration reflects a hormetic principle, mobilizing just enough stress response resources to overcome the threat while avoiding unnecessary damage from overactivation [115,127].

#### 8.2.2. Neuroplastic Priming

On a longer timescale, repeated or intermittent exposure to low or moderate levels of CRH appears to condition neural circuits involved in emotion regulation, memory, and autonomic control. This has been demonstrated in the prefrontal cortex, amygdala, and hippocampus, where CRH signaling influences synaptic plasticity, dendritic remodeling, and epigenetic programming [99,128,129]. For example, mild, intermittent CRH signaling enhances cognitive flexibility and emotional regulation, fostering the development of a more resilient brain capable of withstanding future stressors, a core outcome of hormetic adaptation. In contrast, chronic or excessive CRH exposure (as seen in toxic stress or trauma) disrupts this balance, leading to maladaptive plasticity, HPA axis dysregulation, and increased vulnerability to anxiety, depression, and metabolic disease [129].

## 9. CRH, Vasopressin and Oxytocin Interact in Developmental and Behavioral Conditioning

CRH also acts as a developmental cue, particularly in early life, where brief stress exposures can “tune” the HPA axis to become either more resilient or more sensitized, depending on the timing, intensity, and context of CRH activity. We demonstrated that early-life exposure to CRH and OT pathway manipulation causes sex-specific alterations in neuropeptide receptor expression and adult stress reactivity [101], aligning with the idea of CRH as a developmental calibrator of neuroendocrine tone. This is a form of stress inoculation, a hormetic phenomenon whereby early exposure to manageable stress can enhance stress resistance later in life [130]. Moreover, several studies showed that CRH influences behavioral plasticity by modulating fear memory, risk evaluation, and stress coping strategies [128]. It helps encode whether a stressor is novel or familiar, and whether prior responses were successful, thereby conditioning the nervous system’s future behavior.

CRH should not be viewed solely as a reactive hormone for acute stress, but as a hormetic regulator and conditioning molecule. Its role may span the following: acute stress orchestration, signal scaling and titration, long-term neuroplastic adaptation, behavioral programming, and the developmental priming of stress resilience. Therefore, CRH lies at the core of a finely tuned conditioning network, shaping both immediate survival responses and the capacity for long-term recovery, learning, and adaptation.

Furthermore, CRH acts as a behavioral conditioner, shaping coping strategies, novelty detection, and arousal modulation. It helps encode contextual threat and adaptive responses in real time, contributing to an organism’s future behavioral flexibility. This function is especially relevant in intermittent stress paradigms like exposure therapy, fear extinction, and resilience training, where controlled CRH signaling facilitates memory updating and extinction learning [129].

Importantly, CRH also interfaces with cardiovascular adaptation. CRH not only activates the sympathetic nervous system via central autonomic pathways, but also modulates vascular tone and heart rate variability. In hormetic contexts, these effects can enhance cardiovascular efficiency and stress responsiveness. However, persistent CRH elevation, as in PTSD or chronic anxiety, leads to sympathetic overdrive and cardiovascular pathology [127].

Beyond its intrinsic signaling capacity, CRH’s effects are intricately modulated by its interactions with VP and OT, two neuropeptides that share overlapping release pathways and physiological roles. In acute stress states, CRH and VP often act synergistically to enhance ACTH release and drive catabolic mobilization [92]; this synergism between CRH and VP facilitates metabolic mobilization and vigilance [89], while OT functions as an opposing force that downregulates CRH-driven sympathetic arousal and restores affiliative behavior post-threat. This CRH-VP coordination increases sympathetic activity, vascular resistance, and metabolic readiness, aligning with the initial phase of a hormetic response.

In contrast, OT frequently rises in the aftermath of stress, counteracting the effects of both CRH and VP. OT suppresses HPA axis hyperactivity, promotes parasympathetic tone, and modulates limbic circuitry to reduce anxiety and promote social recovery behaviors [6,64,92]. These effects are well supported by our previous study [6], in which we proposed that OT acts as a neurochemical “recovery signal” following CRH-driven arousal, restoring homeostasis and modulating cardiovascular and emotional responses through prefrontal-limbic circuitry. As such, the CRH-OT dynamic can shape both the magnitude of the initial stress response and the quality of post-stress recovery and integration.

These interrelated dynamics suggest that CRH, VP, and OT act not as isolated hormones, but as neuroendocrine instructors in a broader hormetic conditioning network. Their coordinated oscillation fine-tunes immediate responsiveness and longer-term plasticity, enabling physiological systems to better predict, endure, and adapt to environmental fluctuation.

Therefore, CRH is not merely a stress response initiator, but a conditioning signal that calibrates neuroendocrine sensitivity, behavioral adaptation, cardiovascular reactivity, and long-term resilience. Through its interactions with VP and OT, it anchors a hormetic feedback loop that balances catabolic mobilization with anabolic repair and neuroplastic adaptation.

### 9.1. CRH as the Gateway to Neuroendocrine Hormesis

CRH initiates a cascade of metabolic, mitochondrial, behavioral, and epigenetic adaptations that exemplify the hormetic principle. While overactivation is pathological, controlled activation through mild physiological stress enhances resilience, flexibility, and energy efficiency [131]. CRH prepares the system not only for challenge but also for recovery, priming the body and brain for the buffering effects of VP and OT, which complete the hormetic cycle by fostering social connection, emotional stability, and long-term homeostasis [6].

### 9.2. Urocortins in Hormetic Stress Modulation—Bridging Catabolic Initiation and Adaptive Recovery

Urocortins play crucial roles in modulating adaptive responses to physiological stress. Functioning via CRH receptors (CRHR1 and CRHR2), UCNs integrate neuroendocrine signaling with metabolic resilience, mitochondrial adaptation, and cardiovascular protection. Within the hormesis framework, UCNs operate downstream of CRH to fine-tune stress responses, amplifying cellular defenses and promoting recovery after transient metabolic or psychosocial challenges.

#### 9.2.1. Urocortin Subtypes and Functional Roles

UCN 1 binds both CRHR1 and CRHR2, and is implicated in mitochondrial biogenesis, neuroprotection, and metabolic flexibility. It activates PGC-1α, enhancing mitochondrial proliferation and oxidative metabolism [70], and supports glucose uptake and fatty acid oxidation under conditions of fasting or caloric restriction [132]. UCN1 also attenuates oxidative stress and apoptosis in the hippocampus, underscoring its neuroprotective capacity [133].

UCN 2 and 3 preferentially bind CRHR2 and mediate anti-inflammatory, cardioprotective, and metabolic functions. They mitigate myocardial ischemic injury and enhance cardiac performance under stress, suppress pro-inflammatory cytokines, and regulate insulin sensitivity and glucose metabolism, guarding against obesity-linked insulin resistance [134]. Additionally, UCN2/3 selectively bind CRHR2, promote adaptive thermogenesis by stimulating brown adipose tissue (BAT) activity, attenuate myocardial injury, enhance cardiac output, and improve insulin sensitivity [92,111,135].

#### 9.2.2. Hormetic Mechanisms of Urocortins Action

Low-level Urocortins activation enhances mitochondrial function through PGC-1α upregulation and promotes ATP production. UCNs also reduce oxidative stress by upregulating antioxidant enzymes like SOD and glutathione peroxidase [92,132,136], provide vascular protection via eNOS activation, and improve systemic metabolic resilience through enhanced lipid and glucose utilization [134]. However, excessive UCNs signaling can disrupt mitochondrial homeostasis, elevate ROS, and impair electron transport chain efficiency. Chronic overactivation contributes to inflammation, cardiomyocyte hypertrophy, maladaptive remodeling, and HPA axis dysregulation, features linked to depression, anxiety, and metabolic syndrome [70,137].

#### 9.2.3. Urocortins and the Transition to Anabolic Recovery

UCNs, particularly UCN2 and UCN3, complement CRH activity by promoting metabolic and emotional recovery. They activate CRHR2, fostering anti-inflammatory, cardioprotective, and neuroregenerative effects [70]. This mirrors the transition from a catabolic (stressor) to an anabolic (recovery) phase central to the CACH model. Importantly, this recovery phase primes the neuroendocrine environment for the action of OT and VP, which further refine stress responses through social and affiliative buffering mechanisms [6,48,57].

### 9.3. Integration into the Catabolic-Anabolic Cycling Hormesis Model

UCNs function as pivotal mediators of the CACH model. During the catabolic phase, UCNs alongside CRH activate the HPA axis, promote AMPK signaling [138], and facilitate energy mobilization through gluconeogenesis, fatty acid oxidation, and autophagy [61]. These actions build stress resistance and initiate epigenetic reprogramming toward metabolic flexibility [139].

In the anabolic phase, after CRH subsides, mTOR signaling is upregulated to drive cellular repair, glycogen storage, and protein synthesis [140]. Vasopressin and OT emerge during this recovery window to restore homeostasis, supporting emotional resilience, cardiovascular regulation, and social bonding [48,57,81]. UCNs’ activity in this phase also enhances insulin sensitivity, promotes neuroplasticity, and reinforces long-term cognitive and physiological adaptation [131].

### 9.4. Urocortins as Bidirectional Stress Regulators

UCNs bridge the initial catabolic stress activation with anabolic recovery and resilience, demonstrating their key role in the hormetic stress response (Figure 2). Their ability to activate protective pathways, including AMPK, mitochondrial biogenesis, and anti-inflammatory signaling, positions them as essential players in the adaptive stress response [92,136]. However, the same mechanisms can become pathologic under chronic or excessive stimulation, leading to metabolic and neuropsychiatric dysregulation [70,137].

Framing UCNs within the CACH model provides a useful framework for understanding their dual role in stress biology. Their interactions with OT and VP further extend their influence into social, emotional, and intergenerational domains of adaptation [6,46]. Future research may aim to harness these pathways therapeutically, including through lifestyle interventions like exercise and fasting or targeted pharmacologic modulation, to optimize health span, metabolic performance, and resilience to psychological stress [136].

## 10. Preparing the Stage for Oxytocin and Vasopressin

### 10.1. Recovery Adaptation Modulators

OT and VP play crucial, complementary roles during stress recovery. OT dampens CRH-induced amygdala activation [49,141], enhances bonding, and promotes parasympathetic tone [49,81], facilitating emotional healing and social reengagement [142,143]. VP, though typically pro-stress during acute threat [43,55], supports circadian rhythm and social hierarchy maintenance under moderate stress [57,100,144]. Together with CRH and UCNs, the VP-OT system forms a feedback loop guiding transition from arousal to recovery [46,57,78]. OT counters CRH-driven anxiety by reducing amygdala activity [141,144] and enhancing vagal tone [49]. This transition supports healing, social approach, and resilience [46,142]. VP amplifies HPA output during acute stress [43,55] but facilitates adaptive responses like energy conservation, circadian alignment, and social vigilance in familiar or moderate stress [57,100,143]. CRH and UCNs initiate the stress response [55,138], while OT and in some cases VP orchestrate recovery [46,57], emphasizing the hypothesis that resilience can depend more on the repair phase than initial stress activation [78]. Adaptive outcomes depend on stressor intensity, timing, and the OT/VP-mediated recovery [46,57,78,145]. Resilience is reconceptualized as the capacity for adaptive recalibration, not just resistance [48,142].

### 10.2. Integration of Oxytocin and Vasopressin Function

VP initially mobilizes energy and alertness [36,43]; OT later facilitates recovery and plasticity [57,135]. The functions of both are shaped by epigenetics [101,144] and modulate neuroimmune and autonomic functions [6,81,113]. Early-life OT/VP exposure programs lifelong stress responsivity in a sex-specific manner [101,146]. VP promotes ACTH release and HPA activation [43,55,102]. Moderate levels enhance vigilance and metabolism [60], while chronic excess contributes to dysfunction, mood disorders, and gut issues [36,44,84,147,148,149]. The role of VP is biphasic—beneficial under mild stress or in the face of immediate demands, but harmful when prolonged [55,60]. OT blunts prolonged HPA activation [46,57,67], protects cognition [116], and supports stress recovery [46,48]. It preserves hippocampal structure [49,116], enhances social buffering [135], and modulates inflammation and immunity [135,150,151]. OT’s actions span neuroendocrine, immune, and gastrointestinal domains [44,46,152,153], reinforcing its role in resilience. OT is not a panacea, but a neuropeptide with evolutionarily conserved and context-specific physiological effects, comparable to other hypothalamic hormones such as CRH and VP.

### 10.3. Dynamic OT-VP Interplay

OT and VP act in sequence; VP dominates during acute arousal [48,74], OT during recovery [49,135]. Their feedback loop calibrates stress responses; VP primes OT release [135,154], and OT suppresses excessive VP signaling [57,155]. Sex differences are prominent—males show stronger VP responses [156], while females benefit more from OT buffering [81,157,158].

### 10.4. Targeted Clinical Applications of OT–VP Modulation

VP–OT balance has therapeutic implications for PTSD, anxiety, depression, and gut–brain disorders. OT enhances recovery [48,135,142], while VP antagonists reduce hyperarousal [159,160]. OT also promotes metabolic health [161,162], cognitive longevity [135,163], and exercise adaptation [161,164].

### 10.5. Hormesis Blueprint for Resilience

Framing OT and VP within a hormetic framework repositions these peptides not as binary stress mediators but as precision regulators of stress-response plasticity. Their therapeutic utility lies in their capacity to modulate neuroendocrine tone, behavior, and systemic function in a dose-, context-, and state-dependent manner. This paradigm supports the need for a dynamic alternative to traditional fixed-dosing approaches in psychiatry and behavioral medicine, emphasizing individual trajectories, timing, and physiological thresholds. As we have emphasized, OT and VP evolved not merely as stress mediators, but as adaptive neurochemical systems capable of regulating resilience, attachment, and autonomic flexibility across varying stress loads [6]. Together, OT and VP constitute a biphasic stress-regulatory system, in which VP primarily facilitates mobilization and OT promotes repair and recovery [46,135]. When well-regulated, this dynamic pairing enables organisms to convert stress into growth and adaptation, a process that exemplifies hormesis [135,142]. Conversely, the dysregulation of this balance is associated with psychiatric, metabolic, and neurodegenerative pathologies [57,116]. A hormetic blueprint that integrates OT and VP thus redefines stress adaptation as a dynamic, recovery-driven process. The targeted modulation of these peptides, taking into account timing, dose, individual differences, and physiological thresholds, may offer therapeutic benefit across a spectrum of conditions including PTSD, anxiety, depression, gut–brain disorders, metabolic syndrome, cognitive decline, and impaired exercise recovery [81,135,159,160,161,162,163,164,165,166]. This framework invites further investigation into how precision neuroendocrine strategies can optimize resilience and systemic health by leveraging the bidirectional and state-dependent properties of OT–VP signaling Table 4.

## 11. Mechanistic Foundations

### 11.1. Hormetic Biphasic Signaling of Vasopressin and Oxytocin

In summary, both VP and OT demonstrate classic biphasic, dose-dependent properties consistent with the hormetic model, in which low to moderate levels of exposure confer adaptive benefits, while higher or chronic exposure results in adverse effects. These peptides are centrally involved in neuroendocrine coordination during stress and recovery phases, and modulate diverse physiological systems, including emotional regulation, cardiovascular control, mitochondrial function, and immune modulation. At low physiological concentrations, OT and VP enhance stress resilience, promote prosocial behavior, and regulate energy balance. However, at high doses or with prolonged activation, both neuropeptides can shift their effects toward promoting allostatic overload, inflammation, and maladaptive neurobehavioral outcomes [46,167].

OT, in particular, exhibits anxiolytic and anti-inflammatory effects at low doses by modulating HPA axis reactivity and enhancing parasympathetic tone. In contrast, repeated or high-dose administration may cross-activate V1a receptors and increase anxiety, promote aggression, or induce receptor desensitization. Similarly, VP promotes acute cognitive and cardiovascular responsiveness at low levels but exacerbates glucocorticoid resistance, sympathetic overdrive, and systemic oxidative stress when overexpressed or persistently elevated. These patterns of biphasic response align with the principles of hormesis, where moderate stress-induced signaling promotes long-term adaptive capacity, but excess burden tips physiological systems into maladaptation [85].

### 11.2. Neuroendocrine Integration

The coordination of the catabolic–anabolic shift through the dynamic interplay of CRH, UCNs, VP, and OT represents a temporally regulated and receptor-specific orchestration of stress response systems. The HPA axis initiates the catabolic phase, while the HNS (hypothalamic–neurohypophyseal system) participates in both stress amplification and its resolution. During acute stress, CRH is secreted rapidly (within minutes) from parvocellular neurons in the PVN, activating CRHR1 in the anterior pituitary to stimulate ACTH release and consequent adrenal glucocorticoid secretion [139]. Simultaneously, VP, co-released from adjacent PVN neurons, potentiates ACTH release via V1b receptors, especially under sustained or repeated stress. This synergistic CRH–VP action intensifies the early catabolic response, mobilizing energy stores, increasing cardiovascular output, and modulating immune and behavioral systems [83].

The temporal orchestration of stress responses by CRH, VP, UCNs, and OT follows a highly regulated sequence that reflects distinct yet overlapping neuroendocrine roles during the phases of stress initiation, maintenance, and resolution (Figure 3). As the stressor abates, UCNs and OT assume key roles in initiating the anabolic recovery phase. UCNs are expressed later in the temporal sequence and signal predominantly via CRHR2, which is expressed in the heart, gut, brainstem, and limbic circuits. The activation of CRHR2 facilitates the termination of HPA output, enhances parasympathetic activity, promotes cardiovascular and immune recovery temporal oscillation between activation (CRH/VP) and restoration (UCNs/OT), and reflects the core temporal structure of the CACH model (Figure 4), wherein controlled stress exposure followed by recovery supports adaptive homeostasis and stress resilience.

Interestingly, in all mammals, the OT and VP genes are positioned in close proximity in an inverted orientation on the same chromosome. This genomic architecture allows for mutually exclusive transcription within a single nucleus, a feature that may contribute to the reciprocal regulation observed at the functional level. Moreover, gene conversion events between the second exons of the OT and VP genes are frequent, limiting sequence divergence and possibly preserving complementary functions. This suggests strong evolutionary pressure to maintain this hormetic dyad, despite distinct transcriptional regulation.

### 11.3. The Temporal Role of Each Peptide in Initiating or Resolving Stress

The temporal orchestration of stress responses by CRH, VP, UCNs, and OT follows a highly regulated sequence that reflects distinct yet overlapping neuroendocrine roles during the phases of stress initiation, maintenance, and resolution (Figure 3). CRH is the earliest and primary hypothalamic peptide released in response to perceived threat, acting via CRHR1 receptors in the anterior pituitary to initiate ACTH secretion and consequent glucocorticoid release from the adrenal cortex [168]. This acute response rapidly mobilizes energy stores, suppresses non-essential functions, and enhances arousal. VP, co-released with CRH from parvocellular neurons in the PVN, sustains and amplifies ACTH secretion through V1b receptor activation, particularly under chronic or repeated stressors, thereby contributing to HPA axis sensitization [83]. This co-activation phase defines the catabolic window, characterized by elevated glucocorticoids, sympathetic drive, and metabolic breakdown. Further insights into the differential dynamics of acute versus chronic HPA axis activation are provided in [169].

In contrast, UCNs, particularly UCN2 and UCN3, emerge later in the stress sequence, and predominantly act via CRHR2 receptors to facilitate stress resolution and promote physiological recovery. CRHR2 signaling counteracts the excitatory CRHR1-driven responses by dampening HPA axis output, restoring cardiovascular tone, and exerting neuroprotective and anti-inflammatory actions [170,171]. UCN1, which has an affinity for both CRHR1 and CRHR2, appears to function as a regulatory buffer during the transition from stress engagement to recovery, modulating emotional responses in limbic circuits. Meanwhile, OT is released from magnocellular neurons of the hypothalamus, and is temporally aligned with the resolution and anabolic phase of the stress cycle (Figure 4). Acting through OT receptors, OT enhances vagal tone, reduces amygdala activation, promotes prosocial behavior, and stimulates anabolic repair mechanisms such as glucose uptake, anti-inflammatory signaling, and parasympathetic dominance [172,173] (Table 5).

The specificity of these effects is critically shaped by the distribution and G-protein coupling of their respective receptor subtypes. VP signals through three receptors, V1a, V1b, and V2, each with distinct tissue localization and signaling properties. V1a receptors (coupled to Gq proteins) are widely distributed in the vasculature, brain, and liver, mediating vasoconstriction, aggression, and social memory. V1b receptors (also Gq-coupled) are predominantly located in the anterior pituitary and limbic regions, modulating ACTH release and behavioral reactivity. V2 receptors, in contrast, are found in the renal collecting ducts, where they couple to Gs proteins to regulate water reabsorption via cAMP-PKA signaling [174]. These divergent pathways allow VP to engage acute cardiovascular responses (V1a), neuroendocrine amplification (V1b), and systemic fluid balance (V2), depending on stressor type and physiological context.

Similarly, OTRs, which are primarily Gq-coupled, initiate PLC-IP3-mediated Ca^2+^ mobilization, but can also recruit PI3K/Akt, MAPK, and cAMP/PKA pathways in a context- and cell-type-dependent manner. Importantly, OXTRs exhibit region-specific expression in the hypothalamus, amygdala, brainstem, and peripheral tissues such as the heart and GI tract, where they mediate anti-inflammatory, anxiolytic, and parasympathetic functions [175]. Under high ligand concentrations or prolonged stimulation, OT can cross-activate V1a receptors, leading to vasopressor and potentially anxiogenic effects, especially under stress-primed conditions [176]. The receptor coupling profile thus determines whether VP/OT signaling promotes hormetic adaptation through the regulated, pulsatile engagement of survival and recovery pathways, or contributes to pathological allostatic overload through sustained excitotoxic and inflammatory signaling. Collectively, the neuropeptidergic system operates with temporal and receptor precision, coordinating the initiation, amplification, and resolution of stress through CRHR1/2, V1a/b, V2, and OTRs, with functional outcomes that range from catabolic mobilization to anabolic repair. This biphasic and phase-specific organization not only underpins normal stress adaptation, but also provides a mechanistic framework for therapeutic interventions that mimic or modulate these oscillatory dynamics.

## 12. Hormesis and Allostasis

VP and OT serve as critical neuromodulators within the neuroendocrine system, mediating both the amplification of acute stress responses and the facilitation of recovery processes, depending on timing, context, receptor engagement, and exposure history. Their dual actions significantly shape the balance between allostatic load, the cumulative physiological burden of chronic or dysregulated stress, and allostatic resilience, the organism’s capacity to recover from stress while maintaining or enhancing system integrity [173,177].

VP, primarily acting through V1a and V1b receptors, augments hypothalamic–pituitary–adrenal (HPA) axis activity and sympathetic arousal during the catabolic phase of stress. Under acute, time-limited challenges, this facilitates adaptive responses—increased ACTH and glucocorticoid secretion, vasoconstriction, and glucose mobilization [83]. However, under recurrent or chronic stress, persistent VP signaling contributes to elevated allostatic load through glucocorticoid resistance, excessive sympathetic output, impaired negative feedback, and heightened inflammation [178]. In contrast, intermittent or context-appropriate VP release, as observed in circadian regulation or controlled social engagement, can promote stress inoculation and precondition neuroendocrine systems for future perturbations [179]. This aligns with evidence that VP modulates adaptive neuroplasticity in hippocampal and hypothalamic circuits when exposure is moderate and episodic.

OT, in contrast, plays a central role in buffering the consequences of stress and restoring homeostasis. Acting through OT receptors, OT attenuates amygdala activity, reduces HPA reactivity, and enhances vagal tone, key features of allostatic resilience [172]. Importantly, OT also facilitates mitochondrial repair, anti-inflammatory cytokine production, and emotional regulation, especially in social or affiliative contexts. Repeated intermittent OT release, such as that induced by physical touch, social bonding, or moderate physical activity, promotes the adaptive recalibration of the stress system, and may prevent the pathological accumulation of allostatic burden [180,181]. Thus, while VP amplifies stress responses, OT supports system recovery; both peptides can either exacerbate or mitigate allostatic load depending on dose, timing, and receptor context, forming a bidirectional regulatory axis.

### Autonomic Effects of Vasopressin and Oxytocin

A core mechanism through which VP and OT influence resilience lies in their ability to mediate the switch between sympathetic and parasympathetic dominance during the stress recovery cycle. VP supports sympathetic drive, enhancing vasoconstriction and alertness via V1a receptor activity in the central and peripheral nervous system [174]. However, it also contributes to the maintenance of arousal states, which, when unresolved, delay recovery and reinforce chronic stress physiology. OT, conversely, promotes parasympathetic engagement through actions on the nucleus ambiguous and dorsal vagal complex, regulating heart rate variability (HRV), gastrointestinal motility, and cardiovagal tone [182]. Notably, OT’s effects are potentiated in environments that provide psychosocial safety cues, such as social warmth or emotional closeness (Figure 5), highlighting its role in context-sensitive recovery initiation [173].

Moreover, both neuropeptides modulate the timing and extent of autonomic transitions, acting as phase-specific mediators within the CACH framework. During acute stress, VP dominates the autonomic landscape, prolonging sympathetic tone to ensure survival. Upon threat cessation, OT gradually rises, promoting parasympathetic rebound, lowering inflammatory mediators, and facilitating anabolic repair. The dynamic interplay between VP and OT is not merely reciprocal but modulatory, as OT can suppress excessive VP-driven arousal, particularly through the OXTR-mediated inhibition of V1a-induced pathways [176]. Therefore, these neuropeptides do not operate in isolation but as components of a precision-timed feedback network that governs the transition from allostatic activation to resilience-based recalibration.

## 13. Translational and Clinical Potential

Intermittent intranasal OT and selective VP receptor agonists or antagonists are gaining traction as potential hormetic therapeutics capable of enhancing stress resilience, emotional regulation, and cognitive flexibility in vulnerable or clinical populations. Unlike chronic dosing, which can induce receptor desensitization or paradoxical effects, intermittent administration mimics natural pulsatile peptide release and aligns with the biphasic adaptive principle of hormesis [85]. Clinical trials using 24 IU intranasal OT administered 1–3 times per week in patients with PTSD, social anxiety, or ASD have demonstrated improvements in fear extinction, social engagement, and parasympathetic reactivation, particularly when paired with behavioral therapies and a context of safety [173,183]. Similarly, selective V1a receptor agonists have shown promise in enhancing social cognition and working memory in schizophrenia and depression, while V1b antagonists like ABT-436 have been explored for stress-related mood disorders [184,185].

However, the overuse or continuous administration of OT or VP analogs poses notable risks. High or prolonged doses of OT can lead to OTR downregulation, reduced efficacy, and even anxiogenic or VP-like effects, possibly through V1a receptor cross-activation. This is likely to be most problematic in individuals with heightened baseline stress or specific OXTR polymorphisms [167,176]. Similarly, chronic VP stimulation can exacerbate HPA axis dysregulation, hypertension, and emotional rigidity, increasing rather than reducing allostatic load. This underscores the importance of therapeutic intermittency, allowing for receptor resensitization, adaptive feedback, and the proper sequencing of catabolic-anabolic cycles, akin to physical training or caloric restriction models of hormesis.

To gauge the success of such interventions, several biomarkers have been proposed to track hormetic adaptation mediated by OT and VP. Salivary or plasma OT and VP levels, when sampled longitudinally across stress and recovery periods, can reflect pulsatile neuropeptide dynamics [186]. It is worth noting that methodological differences in peptide quantification (e.g., immunoassays vs. mass spectrometry) can significantly affect data interpretation. It is worth noting that methodological differences in peptide quantification (e.g., immunoassays vs. mass spectrometry) can significantly affect data interpretation. The specificity and sensitivity of oxytocin and vasopressin bioassays remain critical issues in the field, particularly in relation to peripheral sampling and behavioral correlations [187].

In addition, heart rate variability (HRV), a well-established index of parasympathetic activity and vagal tone, is positively associated with OT signaling and successful stress recovery. Increases in high-frequency HRV following OT-enhancing interventions (e.g., therapy, exercise, or dosing) signal autonomic rebalancing and neurovisceral integration [182]. Other informative metrics include diurnal cortisol slope, OT receptor gene methylation status, and functional neuroimaging markers, such as improved amygdala–prefrontal coupling and decreased default mode network overactivity.

Together, these biomarkers allow the real-time and long-term assessment of peptide efficacy and resilience building. Given these properties, OT and VP are strong candidates as mediators of stress inoculation strategies, especially in high-risk populations such as caregivers, first responders, military personnel, and older adults. These groups are particularly vulnerable to cumulative allostatic load, emotional exhaustion, and neurodegenerative stress pathways. Recent insights emphasize that central oxytocin deficiency—particularly arising from hypothalamic–pituitary damage—can result in a spectrum of clinical manifestations, including emotional dysregulation, fatigue, and social withdrawal, further underscoring the therapeutic relevance of oxytocin in stress-related and neuroendocrine disorders [188]. Emerging research suggests that targeted OT interventions, in a safe context and coupled with cognitive training, mindfulness-based stress reduction, or exposure therapy, can enhance resilience, interpersonal connection, and physiological recovery in these cohorts [189]. For example, intranasal OT has been shown to enhance affect regulation and increase HRV in caregivers of patients with dementia, suggesting a role in buffering chronic caregiving stress. Similarly, VP receptor antagonism may offer a novel route to reduce over-reactivity in stress-sensitive professions, although careful considerations of context, dose titration and genetic screening (e.g., AVPR1a and AVPR1b SNPs) may be needed to optimize outcomes [178].

Ultimately, these peptides are not universal solutions but phase-sensitive neuromodulators best applied within adaptive stress frameworks, whether clinical or preventive. When embedded within structured recovery cycles, monitored via physiological biomarkers, and matched to individual receptor genotypes or epigenetic profiles, intermittent OT/VP-based therapies may offer a powerful tool to preempt psychiatric disorders and enhance psychological resilience in the face of chronic adversity.

## 14. Cytokines as Dynamic Modulators of Hormetic Responses

Cytokines, long regarded primarily as mediators of host defense and inflammation, have emerged as pivotal regulators of stress adaptation across physiological systems. Within the framework of hormesis, cytokines are not merely end-products of injury or infection, but dynamic signaling molecules that initiate, amplify, and resolve adaptive biological responses [190]. Transient cytokine activation, particularly in response to low-dose or moderate stressors, plays a central role in priming tissues for resilience, enhancing cellular repair capacity, and fostering long-term adaptive plasticity.

Recent findings have significantly redefined the traditional pathogenic view of cytokines. Controlled, time-limited inflammatory responses now appear crucial for stress resilience, synaptic remodeling, mitochondrial health, and metabolic regulation [191]. Cytokines such as interleukin-6 (IL-6), tumor necrosis factor-alpha (TNF-α), and interleukin-1 beta (IL-1β) exhibit context-dependent duality; at moderate levels and appropriate timing, they initiate protective pathways including the activation of nuclear factor erythroid 2–related factor 2 (Nrf2), sirtuin signaling, and heat shock protein (HSP) expression [192].

Importantly, the phase-specific orchestration of cytokine activity governs the catabolic-anabolic transition critical for hormetic success. During the early catabolic phase, moderate pro-inflammatory cytokine production mobilizes energy reserves, initiates cellular defense programs, and activates autophagic and proteostatic systems. Subsequently, the timely rise of anti-inflammatory cytokines supports anabolic recovery, tissue repair, neurogenesis, and immunometabolic reprogramming [193,194].

Emerging research further reveals that hormetic cytokine modulation is intimately connected to mitochondrial stress signaling (mitokines), senescence-associated secretory phenotypes (SASP) under controlled conditions, and epigenetic plasticity that primes cells for future stress encounters [195]. Thus, cytokines serve not merely as transient inflammatory triggers, but as dynamic modulators of cellular memory and organismal resilience.

Failure to properly resolve cytokine-mediated stress responses, through excessive magnitude, prolonged duration, or inappropriate spatial spread, results in the breakdown of hormetic processes and the promotion of chronic inflammatory pathologies, such as metabolic syndrome, neurodegeneration, cardiovascular disease, and cancer [196]. Therefore, cytokines represent both a necessary initiator and a potential saboteur of hormetic adaptation, depending on the precision of their regulation.

In sum, cytokines occupy a central node in the hormesis model, bridging immediate stress responses with long-term adaptations by regulating the transition from catabolic mobilization to anabolic recovery. Their temporal, spatial, and quantitative control mechanisms represent essential targets for therapeutic strategies aiming to enhance resilience, optimize stress inoculation, and prevent maladaptive aging processes.

### 14.1. Pro-Inflammatory Cytokine Surge: The “Alarm Phase” of Hormesis

Upon exposure to low-to-moderate intensity stressors, such as heat shock, physical exertion, ischemic preconditioning, intermittent fasting, or subclinical infection, a rapid but transient surge of pro-inflammatory cytokines initiates the earliest phase of hormesis, often termed the “alarm phase” [190]. Rather than representing maladaptive inflammation, this initial cytokine burst serves as an evolutionarily conserved danger signal that mobilizes broad-spectrum cellular defenses. Key cytokines driving this phase include the following: Tumor necrosis factor-alpha (TNF-α) [197], Interleukin-1 beta [198], and Interleukin-6 [199]. Importantly, the hormetic benefit depends on the magnitude, duration, and resolution of the cytokine surge. Persistent or excessive inflammatory signaling, in contrast, transitions into chronic inflammation, oxidative damage, and tissue pathology, a maladaptive outcome avoided by proper hormetic regulation [200].

Thus, the tightly orchestrated pro-inflammatory cytokine surge during the alarm phase represents a fundamental, beneficial stress-priming mechanism critical for initiating cytoprotective, reparative, and adaptive pathways central to the hormetic response.

### 14.2. Transition Phase: Activation of Cytoprotective and Repair Programs

Following the initial alarm phase, when stress exposure persists but remains within a hormetic intensity window, a critical transition occurs wherein cytokine signaling shifts from acute inflammatory mobilization to adaptive, cytoprotective reinforcement. This phase is essential for facilitating tissue repair, metabolic recalibration, and long-term resilience, rather than progressing toward pathology [200].

This hormetically tuned cytokine signaling allows stressor to promote resilience rather than degeneration. It bridges the catabolic mobilization of the alarm phase to the anabolic recovery phase, establishing a foundation for adaptive plasticity, metabolic reserve enhancement, and immunological recalibration [201].

Failure to appropriately transition, due to excessive stressor magnitude, persistent cytokine activation, or impaired resolution mechanisms, leads to maladaptive chronic inflammation, mitochondrial dysfunction, and disease progression rather than hormetic benefit.

### 14.3. Resolution Phase: Anti-Inflammatory and Anabolic Signaling

A critical feature distinguishing beneficial hormetic responses from pathological outcomes is the efficient resolution of the initial inflammatory response. Following the early surge in pro-inflammatory cytokines, a timely shift toward anti-inflammatory and regenerative signaling is indispensable to restore tissue homeostasis, repair damage, and promote adaptation.

### 14.4. Failure of Resolution: When Hormesis Becomes Pathology

Although transient pro-inflammatory activation is essential for initiating hormetic adaptation, failure to resolve the inflammatory response transforms hormesis from a protective into a pathological process. When cytokine signaling is excessive, prolonged, or dysregulated, multiple maladaptive consequences emerge.

### 14.5. Biological and Clinical Implications of Cytokine Surges

The success of hormetic adaptation hinges not only on initiating appropriate inflammatory responses, but critically on their timely resolution.

Strategies to modulate cytokine surges, enhance anti-inflammatory pathways, and optimize stress dosing are crucial to prevent hormesis failure and reduce disease burden Figure 6.

### 14.6. Neuroimmune Interactions: OT, VP, and Cytokine Modulation

The intricate integration between cytokine networks and neuropeptides, particularly OT and VP, has emerged as a pivotal axis in orchestrating successful hormetic adaptation. Rather than acting solely within the nervous or immune systems, these peptides function as neuroimmune regulators, modulating inflammation, stress recovery, and cellular resilience across the catabolic-to-anabolic continuum.

#### 14.6.1. Oxytocin: A Neuroimmune Brake on Inflammation

Direct Inhibition of Pro-inflammatory Cytokines. OT exerts potent anti-inflammatory effects by directly suppressing the secretion of TNF-α, IL-1β, and IL-6 from macrophages, monocytes, and microglia [202]. OT also enhances parasympathetic (Vagal) Tone. OT may strengthen the vagal anti-inflammatory reflex by acting on brainstem nuclei (e.g., dorsal vagal complex), supporting a protective mechanism against excess inflammation.

#### 14.6.2. Neuroprotection via Microglial Modulation

In the CNS, OT reduces microglial activation, inhibits NLRP3 inflammasome formation, and prevents the release of neurotoxic cytokines such as IL-1β and TNF-α [173]. This microglial quiescence protects against stress-induced synaptic loss, hippocampal atrophy, and cognitive dysfunction during and after hormetic stressors (Table 6).

## 15. Integrated Neuroimmune Crosstalk

The balance between OT and VP activity critically tunes cytokine dynamics across the hormetic response. Effective hormetic adaptation depends on phase-appropriate switching: VP-driven mobilization during the alarm and early catabolic stages must be followed by OT-mediated resolution, regeneration, and stress recalibration. Failure to achieve this balance, due to excessive VP or insufficient OT activity, can derail hormesis into chronic inflammation, emotional dysregulation, and accelerated aging. All of these are features of disorders such as severe PTSD, for which an imbalance of VP and OT have recently been documented [18].

## 16. Future Directions and Research Priorities

Key questions for advancing the clinical application of CACH include the following: What are the optimal oscillation frequencies and recovery intervals to induce maximal benefit without promoting maladaptation? How does biological age, sex, or disease state affect CACH dynamics and responsiveness? Can digital biomarkers (e.g., heart rate variability, mitochondrial respiration, neuropeptide levels) be used to monitor real-time engagement of CACH? Can neuropeptide analogues (e.g., VP/OT modulators) be used therapeutically to entertain beneficial cycles in aging or cardiovascular disease? Can awareness of the hormetic effects of OT and VP be used to index the effectiveness of therapeutic approaches, including those involving either physiology or behavior?

The CACH model provides a powerful, temporally structured lens through which to understand the biological adaptation to stress. By coupling stress-induced activation with recovery-driven repair, this model unifies concepts from mitochondrial biology, neuroendocrinology, and systems physiology. Central to its orchestration are VP and OT, whose cyclical roles in stress and recovery represent a highly conserved mechanism for enhancing resilience, optimizing performance, and preventing chronic disease. As precision medicine shifts toward resilience engineering, CACH offers a conceptual blueprint for designing therapies that harmonize with the body’s natural rhythms of adaptation (Table 7).

## 17. Conclusions

The integration of VP and OT into the hormetic framework provides a paradigm-shifting perspective on how ancient neuropeptides mediate the dynamic oscillation between stress activation and recovery, catabolism and anabolism, vulnerability and resilience. These molecules, previously understood as key modulators of social behavior and homeostasis, are now recognized as core components of a neuroendocrine circuitry that orchestrates systemic adaptation across multiple physiological domains, including immune regulation, metabolism, emotional processing, and cognitive flexibility.

Conceptualized through the lens of CACH, VP and OT serve as temporal gatekeepers of biological rhythm. VP facilitates acute stress reactivity via HPA axis activation, sympathetic tone augmentation, and defensive behaviors, thereby enabling short-term survival and energy mobilization. In contrast, OT mediates reparative processes by downregulating excess arousal and anxiety, enhances parasympathetic balance, modulates inflammation, and promotes social and emotional reintegration. This sequential interplay reflects a fundamental principle of hormesis—beneficial adaptation is contingent upon the successful transition from activation to recovery, and from disruption to recalibration.

Importantly, VP and OT do not act in isolation. They function within a broader network of regulatory peptides, including CRH and UCNs, which initiate, sustain, or resolve stress responses depending on the intensity, duration, and contextual framing of the stressor. CRH and UCNs activate catabolic signaling and neuroendocrine alertness, particularly via CRHR1 and CRHR2, while OT and VP modulate feedback inhibition and downstream repair pathways. The integration of these systems reveals a multistage, homodynamic model in which neuropeptide interactions encode both the threat appraisal and adaptive outcome of a stressor.

Critically, the developmental and sex-specific regulation of these neuropeptide systems must be acknowledged. Early-life adversity, attachment quality, social buffering and perceived safety calibrate lifelong VP and OT tone, with implications for stress susceptibility and resilience. Males and females exhibit differential sensitivity to VP- and OT-mediated effects, suggesting that personalized interventions should account for sex-based neuroendocrine phenotypes and their plasticity across the lifespan.

Therefore, we reemphasize the concept that VP and OT represent more than stress or “anti-stress” hormones. VP and OT serve as evolutionarily conserved mediators of physiological flexibility, social regulation, and adaptive plasticity. It is of interest that CRH, the UCNs and cytokines all existed before the evolution of the modern VP-OT system [14]. We specifically hypothesize here that in mammals, the VP-OT system may have a hierarchical capacity to coordinate these older systems with the behavioral demands of contemporary social species.

The orchestrated, context-dependent, and reciprocal dynamics of these interactive systems offer a unifying framework for understanding and therapeutically harnessing the process identified hormesis. By advancing this integrative neuroendocrine model, we hope to suggest new avenues for precision stress medicine and resilience engineering across mental and physical health domains.

## Figures and Tables

**Figure 1 cimb-47-00632-f001:**
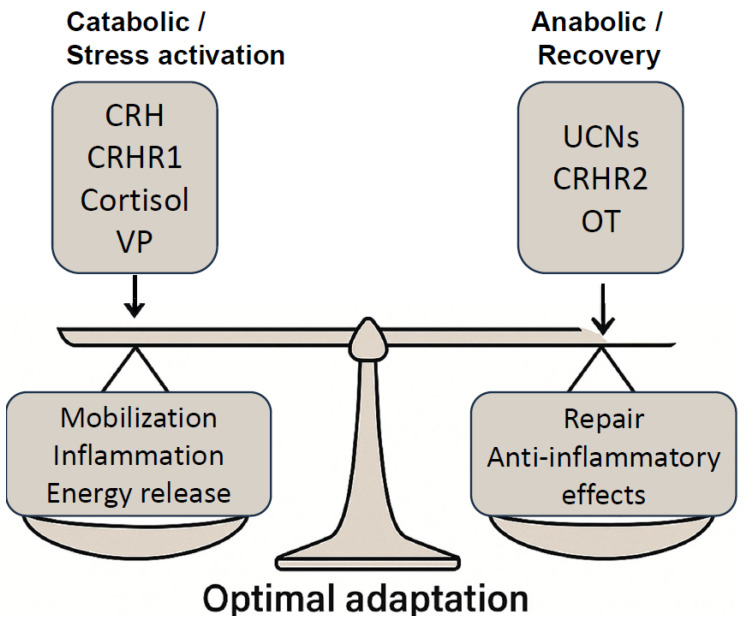
Hormesis represents the dynamic balance between catabolic stress activation and anabolic recovery, orchestrating optimal physiological adaptation. During the catabolic phase, mediators such as cortisol, CRH, VP, and CRHR1 signaling promote mobilization, inflammation, and energy release necessary for immediate stress responses. In contrast, the anabolic recovery phase is governed by OT, UCNs, and CRHR2 pathways, which facilitate tissue repair and exert anti-inflammatory effects. Maintaining equilibrium between these opposing forces enables the organism to maximize resilience and adaptability, highlighting hormesis as a critical framework for health optimization.

**Figure 2 cimb-47-00632-f002:**
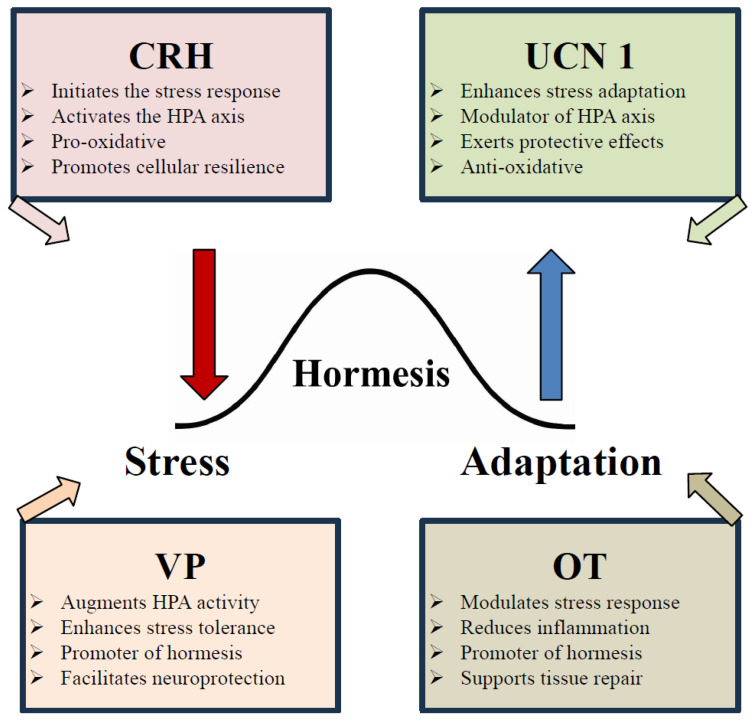
The regulation of hormesis is orchestrated by the dynamic interplay of CRH, UCN1, VP, and OT. CRH initiates the stress response through HPA axis activation, promoting a pro-oxidative state that primes cellular resilience. In contrast, UCN1 enhances stress adaptation by exerting protective, anti-oxidative effects and modulating the HPA axis. VP acts as a potent promoter of hormesis, augmenting stress tolerance, neuroprotection, and HPA axis activity. Meanwhile, OT facilitates recovery by modulating the stress response, reducing inflammation, and supporting tissue repair.

**Figure 3 cimb-47-00632-f003:**
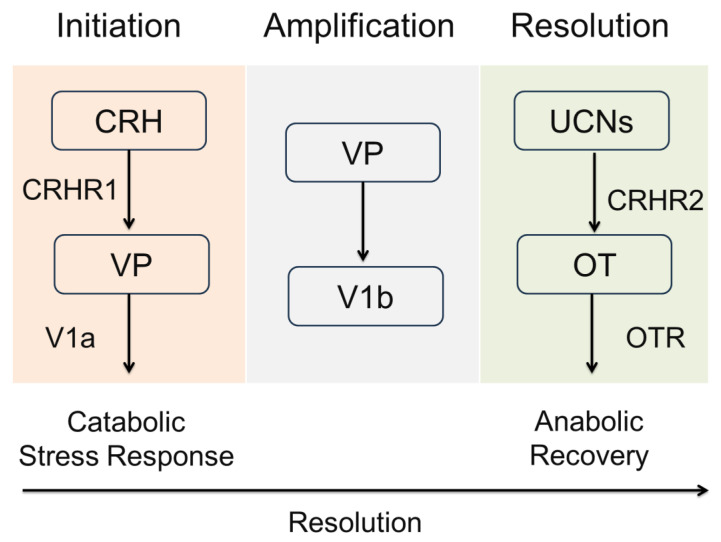
The temporal orchestration of stress responses by CRH, VP, UCNs, and OT follows a highly regulated sequence that reflects distinct yet overlapping neuroendocrine roles during the phases of stress initiation, maintenance, and resolution. CRH, UCNs, VP, and OT interact dynamically to coordinate the shift from catabolic stress responses to anabolic recovery.

**Figure 4 cimb-47-00632-f004:**
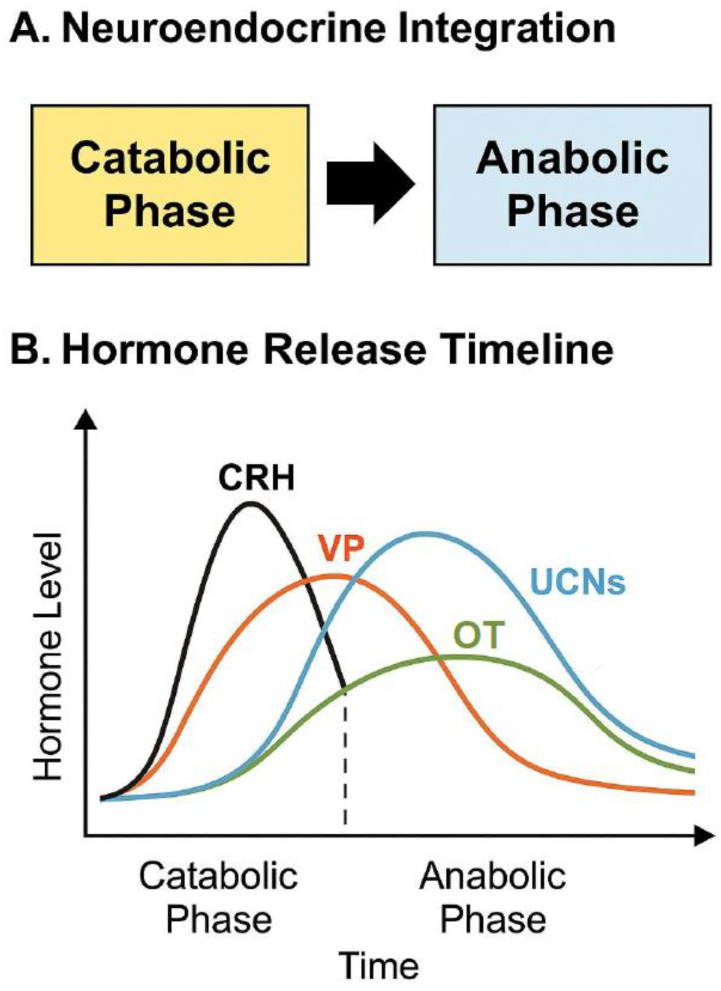
Neuroendocrine integration across stress phases. (**A**) Schematic representation of the biphasic neuroendocrine response to stress, transitioning from a catabolic phase (yellow) to an anabolic phase (blue), reflecting dynamic endocrine and metabolic adaptation. (**B**) Hormone release timeline showing relative levels of CRH (black), VP (red), UCNs (blue), and OT (green) across stress phases. CRH peaks early during the catabolic phase, followed by VP, which bridges both phases. UCNs and OT rise predominantly in the anabolic phase, supporting recovery, regeneration, and recalibration. Hormonal transitions are temporally aligned with physiological shifts from mobilization to restoration.

**Figure 5 cimb-47-00632-f005:**
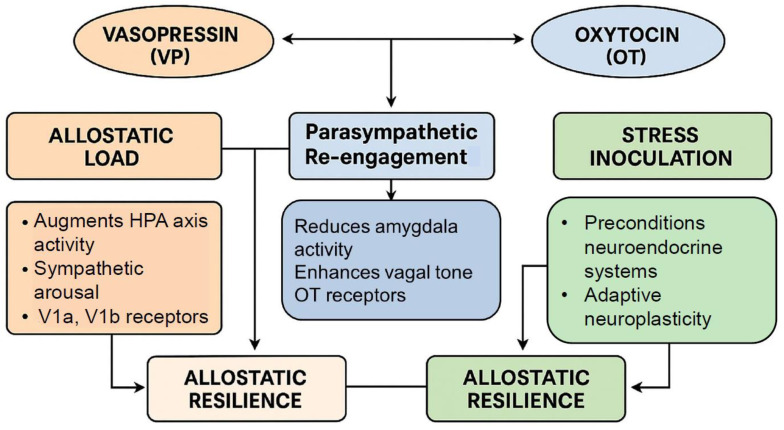
Vasopressin and oxytocin exert distinct but coordinated influences on stress adaptation. VP primarily promotes allostatic load by augmenting HPA axis activity and sympathetic arousal through V1a and V1b receptors, while OT facilitates stress inoculation by enhancing adaptive neuroplasticity and preconditioning neuroendocrine systems. Both peptides converge on parasympathetic re-engagement mechanisms—reducing amygdala activity and enhancing vagal tone—to ultimately promote allostatic resilience.

**Figure 6 cimb-47-00632-f006:**
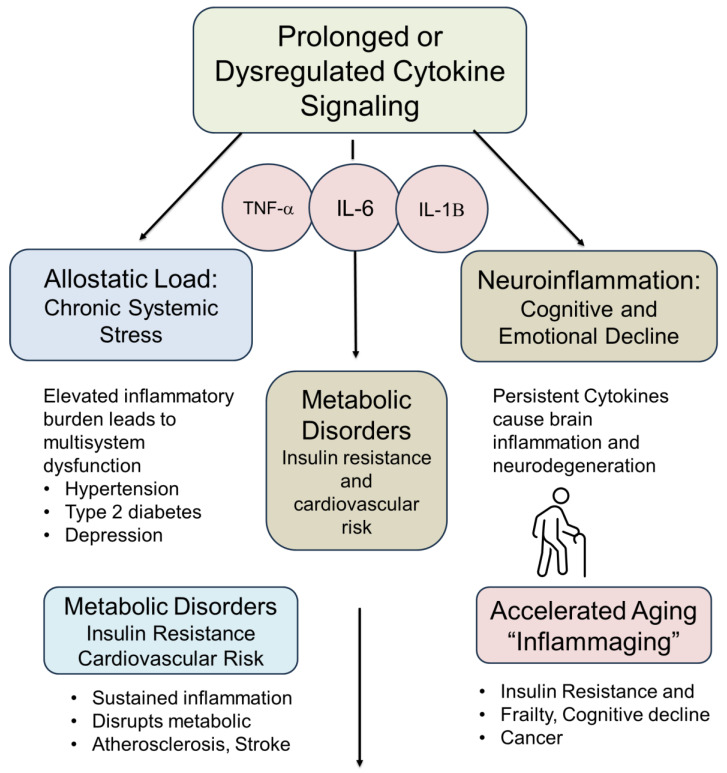
Failure of resolution: transition from hormesis to pathology. This diagram illustrates how prolonged or dysregulated cytokine signaling—specifically chronic elevation of TNF-α, IL-6, and IL-1β—leads to the breakdown of hormetic adaptation and drives pathological outcomes. Failure to resolve early inflammatory activation results in four major maladaptive cascades: (1) Allostatic load, characterized by systemic stress and promoting hypertension, type 2 diabetes, and depression; (2) neuroinflammation, leading to hippocampal atrophy, cognitive decline, and increased neurodegenerative risk; (3) metabolic disorders, including insulin resistance, atherosclerosis, and cardiovascular disease; and (4) accelerated aging (“Inflammaging”), associated with frailty, cognitive impairment, and cancer. Successful hormesis requires the precise regulation and timely resolution of inflammatory responses to prevent these maladaptive trajectories.

**Table 1 cimb-47-00632-t001:** This table outlines key conceptual domains and unresolved research questions regarding the roles of oxytocin (OT) and vasopressin (VP) in stress regulation. It organizes future directions across four tiers—mechanistic foundations, neuroendocrine integration, adaptive stress responses (hormesis and allostasis), and translational/clinical potential—highlighting essential gaps in knowledge from molecular mechanisms to therapeutic applications.

Conceptual Domains	Unresolved Research Questions
Mechanistic Foundations	How do OT and VP exhibit biphasic, dose-dependent responses? What intracellular pathways are engaged under low vs. high stress?
Neuroendocrine Integration	How do CRH, UCNs, VP, OT orchestrate the catabolic-anabolic shift? What is the temporal role of each peptide in initiating or resolving stress?
Hormesis and Allostasis	How do VP and OT affect allostatic load vs. resilience? Can they promote parasympathetic re-engagement after acute stress?
Translational and Clinical Potential	What is the optimal timing/dose for clinical use in PTSD, depression, aging? Are there risks of desensitization or dysregulation with chronic exposure?

**Table 3 cimb-47-00632-t003:** Biphasic catabolic → anabolic cycles in key physiological systems. This table shows where catabolic-to-anabolic cycling appears across whole-organism systems. It summarizes how diverse biological systems engage in hormetic stress cycles, where an initial catabolic phase triggered by challenge is followed by an anabolic phase of restoration and adaptation. They reflect the CACH model, highlighting the evolutionary advantage of alternating between stress and recovery. Phase specific molecular drivers of CACH. This table outlines the key molecular regulators that operate during the catabolic and anabolic phases of hormesis. The listed signaling pathways function in a phase-specific manner to coordinate energy use, cellular defense, and tissue regeneration, supporting systemic resilience and adaptive plasticity under fluctuating stress conditions.

System	Catabolic Phase (“Challenge”)	Anabolic Phase (“Restoration”)	Primary References
Skeletal muscle physiology	Myofibrillar micro-trauma produced by resistance exercise ↑ AMPK & ROS signaling; temporary protein breakdown	Fiber hypertrophy; satellite–cell fusion Mitochondrial biogenesis (↑ PGC-1α, mTORC1)	[53]
Metabolic regulation	Lipolysis, ketogenesis, autophagy during caloric restriction/intermittent fasting Activation of SIRT1, FOXO, glucocorticoids	Re-feeding restores glycogen; ↑ insulin sensitivity Protein and lipid synthesis via IGF-1 → Akt → mTORC1	[51]
Cognitive performance/brain	Controlled cognitive load or acute psychological stress → HPA-axis and catecholamine surge Transient synaptic destabilization	Parasympathetic rebound; sleep-dependent synaptic remodeling Neurogenesis and BDNF-mediated circuit strengthening	[61]
**Phase**	**Key Signaling Nodes/Mediators**	**Core Functions**	**Primary References**
Catabolic (Energy-Mobilizing)	AMPK, SIRT1, FOXO—energy conservation, autophagy, antioxidant defense Glucocorticoids, catecholamines, vasopressin—substrate mobilization, cardiovascular tone Sub-toxic ROS → Nrf2, HSPs, UCPs—stress-sensor activation, mitochondrial uncoupling	Maintain vital function under challenge; initiate cellular “clean-up”	[51,53,69]
Anabolic (Recovery/Growth)	mTORC1, IGF-1, Akt—protein synthesis, mitochondrial biogenesis, membrane repair Oxytocin, IL-10, BDNF—anti-inflammatory resolution, synaptic plasticity, tissue re-growth	Restore and enhance structure/function; build resilience to future stressors	[32,33,64]

**Table 4 cimb-47-00632-t004:** Comparative roles of oxytocin and vasopressin in hormetic stress regulation. This table summarizes the distinct yet complementary functions of OT and VP across key dimensions of the stress response within the hormetic framework. Each row highlights a specific physiological or behavioral domain, detailing the contributions of OT and VP to stress initiation, adaptation, and recovery. OT predominantly modulates the anabolic recovery phase, promoting emotional, autonomic, immune, and social recalibration, while VP primarily drives the catabolic activation phase, enhancing arousal, vigilance, and metabolic mobilization. Their dynamic interplay orchestrates resilience, with phase-specific and context-dependent modulation shaping adaptive versus maladaptive outcomes.

Aspect	Oxytocin (OT)	Vasopressin (VP)	References
Primary Role in Stress Response	Promotes recovery, emotional regulation, social bonding, resilience	Initiates acute stress response, vigilance, energy mobilization	[6,46,48,49,81,91,142]
Timing of Activation	Activated during post-stress recovery (delayed response)	Activated immediately during stress (early response)	[46,74,135]
Interaction with CRH	Inhibits CRH-induced amygdala activation; suppresses CRH, ACTH	Potentiates CRH activity and ACTH release	[46,57,141]
HPA Axis Effects	Negative feedback on HPA axis; enhances glucocorticoid receptor sensitivity	Stimulates HPA axis and increases glucocorticoid output	[46,47,57,67,135]
Social Behavior Modulation	Enhances prosocial behavior, trust, and social bonding	Supports dominance, aggression, territorial behavior	[48,81,135,142]
Cognitive Effects	Improves cognitive flexibility, prevents hippocampal atrophy	Enhances memory consolidation, but excessive levels cause anxiety and PTSD	[49,116]
Autonomic Regulation	Enhances parasympathetic tone, reduces sympathetic arousal	Increases sympathetic arousal; may disrupt recovery	[49,81,135]
Gastrointestinal Function	Restores vagal motility, reduces inflammation, supports gut integrity	Regulates motility (via V1a/V1b); excess leads to GI dysfunction	[44,46,152,153]
Immune Modulation	Suppresses pro-inflammatory cytokines; activates T-regs and M2 macrophages	Amplifies inflammation under chronic stress	[135,150,151]
Neuroendocrine Plasticity	Programs stress resilience, especially during development	Modulates HPA tone, stress coping (context-dependent)	[88,101,143]
Sex Differences	Stronger response in females; enhances cardiovascular and social resilience	Stronger in males; linked to aggression and prolonged HPA activation	[81,114,135,157]
Therapeutic Potential	Used in PTSD, depression, GI disorders, aging, and metabolic recovery	Targeted in PTSD, anxiety, schizophrenia; VP antagonists under study	[44,45,81,135,161,164,165,166]

**Table 5 cimb-47-00632-t005:** Temporal dynamics and functional roles of CRH, VP, UCNs, and OT in stress adaptation. CRH initiates the early catabolic phase within seconds, followed by VP sustaining mid-to-late catabolic responses [35,168,169]. UCNs promote transition to anabolic recovery, while OT supports prolonged anabolic repair, social bonding, and anti-inflammatory effects [170,171,172,173].

Hormone	Onset Time	Peak Time	Duration	Phase	Functional Role
CRH	Onset within seconds to 2 min after stress	Peaks at 10–20 min	Returns to baseline by ~60–90 min	Early Catabolic	Initiates HPA axis, stimulates ACTH, mobilizes energy
VP	Onset: 2–10 min	Peaks at 20–40 min	May persist up to 2–4 h, with chronic stress	Mid-to-Late Catabolic	Prolongs ACTH release, enhances cardiovascular and metabolic drive
UCNs (esp. UCN2/UCN3)	Onset: ~30–60 min post-stressor	Peaks at 1–3 h	Sustained up to 6 h	Transition to Anabolic	Dampens HPA activation, promotes neuroprotection and tissue repair
OT	Onset: ~20–60 min post-stressor (delayed)	Peaks at 1–4 h	Sustained release up to 12–24 h (especially in recovery-promoting contexts)	Anabolic	Supports parasympathetic tone, social behavior, anti-inflammation, and regenerative recovery

**Table 6 cimb-47-00632-t006:** Integrated neuroimmune crosstalk between OT and VP in hormetic stress adaptation. This table illustrates the complementary yet opposing roles of oxytocin and vasopressin in regulating cytokine responses, autonomic balance, microglial activation, and stress-phase dominance across the hormetic spectrum. Vasopressin facilitates early stress responses by amplifying pro-inflammatory cytokines, enhancing sympathetic drive, and sustaining glial activation—mechanisms vital for acute mobilization. In contrast, oxytocin supports recovery and resilience by dampening inflammation, promoting vagal tone, and facilitating neuroplasticity. Successful hormesis requires phase-appropriate neuropeptide switching, whereby early VP-driven activation must transition into OT-mediated resolution. Imbalance in this axis, such as prolonged VP signaling or impaired OT release, can disrupt adaptive plasticity and promote chronic pathophysiology including inflammation, affective instability, and premature aging.

Axis	Oxytocin	Vasopressin
Cytokine Modulation	Suppression TNT-a, IL-b, IL-6	Amplifies early pro-inflammatory signals
Parasympathetic/ Sympathetic	Enhances vagal tone, recovery	Boosts sympathetic drive, mobilization
Microglial Activity	Inhibits activation, promotes neuroplasticity	Sustains glial activation if prolonged
Stress Phase	Dominates resolution and recovery phase	Dominates catabolic and alarm phases

**Table 7 cimb-47-00632-t007:** Future directions for advancing CACH research and translational applications. This table presents a strategic roadmap for the future of CACH-oriented research, ranked by translational potential. Longitudinal developmental work is essential for informing early prevention, but is longer-term in scope.

Priority	Future Direction	Focus Area	Potential Impact	Notes
1	Multi-omic profiling to map individual stress-response signatures	Epigenomics, transcriptomics, proteomics	Personalized hormesis models; identification of hormetic thresholds and maladaptive tipping points	Critical for precision medicine and individualized resilience protocols
2	Neuroadaptive technologies for real-time modulation of resilience circuits	Real-time fMRI, tDCS/TMS, closed-loop OT/VP delivery	Enables state-contingent intervention; rapid feedback-based enhancement of stress recovery and cognitive performance	High innovation; bridges neuroscience and technology
3	Integrated lifestyle-based hormetic interventions	Combined use of CR, exercise, social bonding, cognitive stress, and neuropeptide enhancement	Scalable, low-cost strategies to increase population-level resilience and healthspan	High feasibility and translatability to clinical and public health settings
4	Longitudinal developmental studies on neuropeptide plasticity	Developmental biology, early life stress, OT/VP programming	Insights into critical windows, reversibility of early adversity, and preventive strategies	Long timeline; foundational for understanding life-course effects

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
