# Peer review of "Oxytocin, Vasopressin and Stress: A Hormetic Perspective"

_cimb, 2025, doi:10.3390/cimb47080632_

Round 1
Reviewer 1 Report
Comments and Suggestions for Authors
This is a super-good review of an extremely complicated and rapidly growing transdisciplinary field. It covers a large number of references and is likely to become a classical review with potentially the role that Mason´s volume of Psychosomatic Medicine had in the development of the HPA-HPG hypothesis. We now have fMRI and a growing number of peptides and other molecules that can help us explain complex phenomena, with the emphasis on regulatory processes with short- and long-term perspectives stretching even into epigenetics and mitochondria. What I like in particular is the cautiousness with which the authors discuss potential conclusions regarding therapies. Introduction of intranasal oxytocin administration is one good example which is discussed.
I do not like the title. Vasopressin should be in the title. …mysterious healing powers of stress… is a flirty kind of title which does not correspond to the serious tone of this review. In fact, the authors never make any sweeping statement about stress as healing. Instead they emphasize that under certain conditions energy mobilization in which vasopressin plays an important role can be beneficial for restoring healthy regulatory mechanisms but that does not correspond to the title. I would like a more boring title.
The figures and tables as well as the language are excellent.
A couple of comments that the authors could take or leave :
I had expected that there would be something about music since music researchers are struck by the ability of oxytocin to influence music experiences and skills (rhythm) linked to social aspects of music listening and making. I have been thinking about that as an educational example of social interaction. (Gebauer et al 2016). Perhaps not surprising given the nasal oxytocin - shyness research but perhaps more exact.
A note regarding bioassays of vasopressin and oxytocin could have been helpful (see Mac Lean 2019). The method of analyzing these hormones is central.
Author Response
Reviewer 1:
Response to Reviewer's Comments
We sincerely thank the reviewer for their exceptionally thoughtful, generous, and encouraging comments. Your comparison of our work to the foundational contributions of Mason in Psychosomatic Medicine is both humbling and deeply appreciated. We are especially grateful for your recognition of the interdisciplinary scope of the manuscript, and your emphasis on our cautious approach regarding therapeutic interpretations, including intranasal oxytocin.
Below we address your specific suggestions in detail:
Title Revision (Inclusion of Vasopressin; Tone Adjustment): We fully agree that vasopressin plays a central and essential role in our review. To better reflect the scientific content and avoid ambiguity, we have revised the title to:
[“Oxytocin, Vasopressin and Stress: A Hormatic Perspective”]
This change emphasizes both neuropeptides and aligns the tone more closely with the scientific depth of the review. We appreciate your guidance in steering us away from overly interpretive language.
Music and Oxytocin: Thank you for this insightful suggestion. We have now included a brief note on the intersection of oxytocin and music-related social experiences. Specifically, we refer to the work of Gebauer et al. (2016), which highlights the role of oxytocin in modulating musical reward and social synchrony. We agree that this domain represents a compelling example of socially mediated oxytocin release and have framed it as a useful illustration of adaptive social engagement in a hormetic context.
[Added text in Section 6.5.3: “For instance, music-based social interactions—such as group singing or rhythmic coordination—have been shown to stimulate oxytocinergic activity and enhance prosocial bonding [114], supporting the idea of ‘social hormesis’ through shared sensory and emotional experience.”]
Methodological Note on OT/VP Bioassays: We appreciate your suggestion to mention the challenges surrounding oxytocin and vasopressin quantification. In response, we have added a short paragraph referencing MacLean et al. (2019), highlighting the importance of assay specificity, sample extraction, and validation in neuropeptide measurement. This addition underscores the technical complexity involved in interpreting circulating hormone levels in translational studies.
[Added text in Section 13: “It is worth noting that methodological differences in peptide quantification (e.g., immunoassays vs. mass spectrometry) can significantly affect data interpretation. As discussed by MacLean and colleagues [189], the specificity and sensitivity of oxytocin and vasopressin bioassays remain critical issues in the field, particularly in relation to peripheral sampling and behavioral correlations.”]
Once again, we thank the reviewer for their generous support and invaluable insights. Your comments have helped us refine and elevate the quality of this manuscript, and we are grateful for the opportunity to improve our work in response.
Sincerely,
Hans P. Nazarloo, M.D., Ph.D. (on behalf of all co-authors)
Corresponding Author
Reviewer 2 Report
Comments and Suggestions for Authors
This is an interesting review of a field which for many is not particularly easy to grasp. Much appears relatively speculative, though in total it presents a quite convincing hypothesis. In fact, it might be better to include the term “hypothesis” or “hypothetical” in the title. Much of the text appears repetitive, though I suspect that the repeats of certain sentences or concepts are necessary in order to bring across difficult concepts.
Fundamentally, I enjoyed this review and do not wish to appear critical. This should become a document for much discussion and further speculation. My only real criticism is that at times it appears somewhat tautological, particularly where it appears to be justifying the notion of an oxytocin-vasopressin dyad in evolutionary terms, where there is very little information to suggest an evolutionary context. We have no real information at all about such concepts as are discussed in animals other than rodents or humans; everything about “lower” species is simply assumed because they have to be older. I was irritated by the early phrasing on line 68 that “the comparatively modern evolutionary origin of OT may explain ...”. And then later on there is considerable weight attached to the notion of an ancient OT-VP dyad. This is confusing, especially since OT or mesotocin or its equivalents are represented at least in lower vertebrates, and the genomic duplication that gave rise to the OT and VP genes occured quite early in animal evolution.
In fact, genomic aspects might have helped some of the arguments. It is an odd phenomenon that in all mammals the OT and VP genes are exceptionally close together in an inverted orientation such that, in any one cell nucleus, transcription will only allow either VP gene expression or OT gene expression but not both simultaneously. The closeness of the two genes to one another also encourages the continual gene conversion events between the second exons of the two genes, and vice versa might itself be a mechanism preventing the physical divergence of the two genes. This is a very old occurrence and offers a further genetic argument that might support their hypothesis that these two genes function hormetically in a “ying-yang” fashion. There is obvious selection pressure supporting this strange mechanism, even though the promoter regions of each gene imply quite discrete and different transcriptional regulation.
Altogether, this is a valuable article, representing a novel and interesting new hypothesis, which I for one should like to see as a focus for discussion among endocrinologists and neuroscientists everywhere. But it is long, and at times not very accessible for the non-specialist. A general shortening and possible clarification of certain points would be beneficial.
Author Response
Response to Reviewer's Comments
We are very grateful for your thoughtful and constructive review. Your comments were not only insightful but also generous in tone, and we are encouraged by your recognition of the value of the review as a hypothesis-generating work that may stimulate broader discussion across disciplines. We address your specific comments below:
Speculativeness, Hypothesis Framing: Thank you for acknowledging the speculative nature of the manuscript while recognizing the strength of the overarching hypothesis. We agree that it would be helpful to signal the theoretical nature of the paper more explicitly. In response, we have revised the title to:
“Oxytocin, Vasopressin and Stress: A Hormatic Perspective”
This change helps set expectations for the reader and appropriately frames the manuscript as a model-building effort intended to stimulate further research.
On Evolutionary Framing and the Oxytocin–Vasopressin Dyad: We appreciate your critique regarding the evolutionary framing of the OT-VP system. You are correct in pointing out the potential confusion between our reference to the “comparatively modern” emergence of OT and the later emphasis on an “ancient dyad.” We have revised the text to clarify this point. Specifically, we now explicitly state that the oxytocin-like and vasopressin-like peptides are both ancient, with gene duplication events dating back to early vertebrates. The sentence “However, as argued elsewhere [4, 14] the comparatively modern, evolutionary origins of OT may explain some of the powerful capacity of OT and positive social experiences to protect and heal.” has been removed and replaced with language that emphasizes functional divergence over evolutionary time, while avoiding implications that OT is newly evolved in a phylogenetic sense. We also now more clearly state the limitations of extrapolating from mammalian and rodent models to other taxa and have tempered speculative language to acknowledge the restricted comparative data available from non-mammalian species.
[Text added in Section 1.1: “Although both oxytocin-like and vasopressin-like peptides emerged from a shared ancestral gene early in vertebrate evolution, our current understanding of their functional specialization is based largely on mammalian and rodent models. While we use evolutionary terminology to highlight adaptive roles, we acknowledge that direct functional evidence from lower vertebrates and invertebrates remains limited.”]
On Genomic Organization and Evolutionary Constraint: This is an excellent suggestion, and we thank you for raising it. We agree that the genomic proximity and orientation of the OT and VP genes—and the evidence for selective pressure maintaining this configuration—provide compelling support for the hypothesis of a co-evolved, functionally interdependent system.
We have now incorporated this genetic dimension into our revised manuscript, including discussion of:
[Added in Section 11.2: “Interestingly, in all mammals, the OT and VP genes are positioned in close proximity in an inverted orientation on the same chromosome. This genomic architecture allows for mutually exclusive transcription within a single nucleus, a feature that may contribute to the reciprocal regulation observed at the functional level. Moreover, gene conversion events between the second exons of the OT and VP genes are frequent, limiting sequence divergence and possibly preserving complementary functions. This suggests strong evolutionary pressure to maintain this hormetic dyad, despite distinct transcriptional regulation.”]
Your insight helped us frame a molecular mechanism that supports the proposed functional interplay of OT and VP, and we are grateful for the opportunity to highlight this underappreciated genomic relationship.
On Length, Accessibility, and Redundancy: Thank you for your candid remarks regarding the manuscript’s length and occasional repetitiveness. We recognize that the density and complexity of the material, particularly for readers outside the field, can reduce accessibility. While some repetition is intentional to reinforce key concepts (e.g., the biphasic stress-response model, or the complementary roles of OT and VP), we have now carefully edited the manuscript for clarity, tightened redundant passages, and consolidated overlapping paragraphs.
Once again, we thank you for your encouraging review and the thought-provoking suggestions, which have substantially improved the clarity, coherence, and rigor of our manuscript.
Sincerely,
Hans P. Nazarloo, M.D., Ph.D. (on behalf of all co-authors)
Corresponding Author